

# CCN estimations at a high-altitude remote site: role of organic aerosol variability and hygroscopicity

Fernando Rejano[1,2], Andrea Casans[1,3], Marta Via[4], Juan Andrés Casquero-Vera[1,3], Sonia Castillo[1,3], Hassan Lyamani[5], Alberto Cazorla[1,3], Elisabeth Andrews[6,7], Daniel Pérez-Ramírez[1,3], Andrés Alastuey[4], Francisco Javier Gómez-Moreno[8], Lucas Alados-Arboledas[1,3], Francisco José Olmo[1,3] and Gloria Titos[1,3]

[1] Andalusian Institute for Earth System Research, IISTA-CEAMA, University of Granada, Junta de Andalucía, Granada, 18006, Spain
[2] GRASP-SAS, Remote Sensing developments, LOA/Université de Lille-1, Villeneuve, D'Ascq, 59655, France
[3] Department of Applied Physics, University of Granada, Granada 18071, Spain
[4] Institute of Environmental Assessment and Water Research (IDAEA-CSIC), Barcelona, 28008, Spain
[5] Applied Physics I Department, University of Malaga, Malaga, 29071, Spain
[6] Global Monitoring Laboratory, National Oceanic and Atmospheric Administration, Boulder, CO, 80305, United States
[7] Cooperative Institute for Research in Environmental Studies (CIRES), University of Colorado, Boulder, CO, 80309, United States
[8] Department of Environment, CIEMAT, Madrid, 28040, Spain

*Correspondence to*: Fernando Rejano (frejano@ugr.es)

**Abstract.** High-altitude remote sites are unique places to study aerosol-cloud interactions since they are located at the altitude where clouds may form. At these remote sites, organic aerosols (OA) are the main constituents of the overall aerosol population, playing a crucial role in defining aerosol hygroscopicity ($\kappa$). To estimate the CCN budget at OA dominated sites, it is crucial to accurately characterize OA hygroscopicity ($\kappa_{OA}$) and how its temporal variability affects the CCN activity of the aerosol population since $\kappa_{OA}$ is not well established due to complex nature of ambient OA. In this study, we performed CCN closures at a high-altitude remote site during summer season to investigate the role of $\kappa_{OA}$ in predicting CCN concentrations under different atmospheric conditions. In addition, we performed an OA source apportionment using Positive Matrix Factorization (PMF). Three OA factors were identified from the PMF analysis: hydrocarbon-like OA (HOA), less-oxidized oxygenated OA (LO-OOA) and more-oxidized oxygenated OA (MO-OOA), with average contributions of 5%, 36% and 59% of the total OA, respectively. This result highlights the predominance of secondary organic aerosol with high degree of oxidation at this high-altitude site. To understand the impact of each OA factor on the overall OA hygroscopicity we defined three $\kappa_{OA}$ schemes that assume different hygroscopicity values for each OA factor. Our results show that the different $\kappa_{OA}$ schemes lead to similar CCN closure results between observations and predictions (slope and correlation ranging between 1.08-1.40 and 0.89-0.94, respectively). However, the predictions were not equally accurate across the day. During nighttime, CCN predictions underestimated observations by 6-16%, while during morning and midday hours, when the aerosol was influenced by vertical transport of particles and/or new particle formation events, CCN concentrations were overestimated by 0-20%. To further evaluate the role of $\kappa_{OA}$ in CCN predictions, we established a new OA scheme that uses the OA oxidation level (parameterized by the $f_{44}$ factor) to calculate $\kappa_{OA}$ and predict CCN. This method also shows a large bias, especially during





midday hours (up to 40%), indicating that diurnal information about the oxygenation degree does not improve CCN

predictions. Finally, we used a neural network model with four inputs: $N_{80}$ (number concentration of particles with diameter >80 nm), OA fraction, $f_{44}$ and surface global radiation) to predict CCN. This model matched the observations better than the previous approaches, with a bias within ±10% and with no daily variation, reproducing the CCN variability along the day. Therefore, neural network models seem to be an appropriate tool to estimate CCN concentrations using ancillary parameters. accordingly.

## 1 Introduction


Cloud condensation nuclei (CCN) are those aerosol particles that act as the seeds for cloud droplet activation. The number of CCN in the atmosphere determines the number of cloud droplets that form. This, in turn, affects cloud properties such as reflectivity and lifetime (Twomey, 1977; Albrecht, 1989), playing a critical role in the regulation of Earth's energy balance, climate and hydrological cycle (Lohmann and Feichter, 2005).

The radiative forcing associated with the indirect effect of aerosols through aerosol-cloud interaction is larger (–1.0 ± 0.7 W m$^{-2}$) than the direct effect of aerosol through aerosol–radiation interaction (–0.25 ± 0.25 W m$^{-2}$) (IPCC, 2023). Therefore, understanding physicochemical properties of aerosol particles that can act as CCN could minimize CCN prediction errors, which are essential to reduce the global aerosol-cloud interactions uncertainty (Seinfeld et al., 2016). For that reason, the spatial and temporal variation of CCN together with parameters controlling CCN concentrations have been studied intensively in the

last decades around the world (Deng et al., 2018; Paramonov et al., 2015; Rose et al., 2010; Salma et al., 2021; Schmale et al., 2018; Park et al., 2023; Kulkarni et al., 2023; Rejano et al., 2021; Che et al., 2016).

If ambient conditions that regulate water vapor supersaturation (SS) are disregarded, the main aerosol properties influencing the CCN activity are particle size, chemical composition and mixing state (Dusek et al., 2006; Cubison et al., 2008; Wang et al., 2010; Deng et al., 2018; Kuang et al., 2020b). To assess how these aerosol properties control the CCN activity under

different ambient aerosol composition and mixing conditions, closure studies (i.e., comprehensive evaluation and comparison of measurements from different instruments or methodologies that aim to measure the same or related parameters) have been proven to be very useful (Cai et al., 2022; Crosbie et al., 2015; Ervens et al., 2010; Jurányi et al., 2010; Ren et al., 2018; Kulkarni et al., 2023).

Particle number size distribution (PNSD) is the main factor controlling CCN estimations (Crosbie et al., 2015; Dusek et al.,

2006). Many studies assume an activation threshold diameter from which all particles are considered activated (Asmi et al., 2011; Cho Cheung et al., 2020; Hoyle et al., 2016; Rose et al., 2017; Casquero-Vera et al., 2023). However, reducing aerosol-cloud interaction uncertainties requires more accurate CCN predictions, which, in turn, requires knowledge about the aerosol chemical composition (Che et al., 2016, 2017).

The effect of chemical composition in CCN activity is usually treated through the hygroscopicity parameter $\kappa$ (Petters and

Kreidenweis, 2007), which can be obtained using bulk or size-resolved chemical composition measurements through a simple



volume mixing rule (Petters and Kreidenweis, 2007). However, while aerosol hygroscopicity of inorganic substances is well characterized, quantification of organic aerosol (OA) hygroscopicity ($\kappa_{OA}$) remains challenging. This is due to the large variety of organic compounds within OA, resulting in a wide range of hygroscopicity values that introduces large uncertainties in CCN predictions (Casans et al., 2023; Hallquist et al., 2009; Jimenez et al., 2009; Zhang et al., 2007). It has been proven that

CCN predictions are very sensitive to $\kappa_{OA}$ and a poor knowledge of $\kappa_{OA}$ variability leads to large biases in CCN closures, especially at OA dominated sites (Cai et al., 2022; Deng et al., 2019; Thalman et al., 2017; Gunthe et al., 2009; Liu and Wang, 2010).

To obtain an accurate estimation of $\kappa_{OA}$, knowledge of OA sources and their time variability are required with high time-resolution (Wu et al., 2016; Deng et al., 2019; Ren et al., 2023; Cai et al., 2018). Positive Matrix Factorization (PMF) has been

proven to be a powerful tool for identifying the main OA components by using the organic mass spectra (Via et al., 2021; Minguillón et al., 2015; Crippa et al., 2013). Previous studies explained $\kappa_{OA}$ variability in terms of OA sources assuming specific hygroscopicity values for each source (Cai et al., 2022; Cerully et al., 2015; Deng et al., 2019; Thalman et al., 2017) or established $\kappa_{OA}$ parameterizations based on the oxidation degree (Duplissy et al., 2011; Mei et al., 2013; Wu et al., 2016; Chen et al., 2017). However, assumptions about $\kappa_{OA}$ needed for accurate CCN predictions vary greatly among studied sites

(Ervens et al., 2010; Cubison et al., 2008; Tao et al., 2021; Kuang et al., 2020b), due to the wide variety of sources and atmospheric processes affecting OA.

Organic aerosol usually dominates aerosol mass concentration in the fine fraction at high-altitude environments (e.g., Fröhlich et al., 2015; Ripoll et al., 2015; Zhang et al., 2023) . In addition, since cloud formation conditions can occur at these sites, high-altitude sites are unique locations for studying aerosol-cloud interactions (Friedman et al., 2013; Li et al., 2020; Iwamoto

et al., 2021; Juráyi et al., 2011). Moreover, these sites are often exposed to free troposphere conditions where the submicron aerosol population tends to be an internal mixture of background particles. In this case, satisfactory CCN predictions can be obtained using simple assumptions about aerosol chemical composition (Juráyi et al., 2010; Duan et al., 2023). However, during some conditions, such as thermally driven upslope flow, high-altitude sites might be influenced by planetary boundary layer (PBL) air with pollution particles being efficiently transported to high-altitude sites and affecting CCN activity

(Jayachandran et al., 2018; Rejano et al., 2021). Also, at these sites high insolation conditions during midday hours promote photochemical processes producing that can lead to new particle formation (NPF) events, completely transforming the background aerosol population from a homogeneous aerosol population to a complex mixture of particles with different chemical and microphysical characteristics (Friedman et al., 2013; Rose et al., 2017; Shang et al., 2018). During these more complex conditions when NPF and/or PBL transport affect the aerosol population, simple approaches for CCN predictions

tend to overpredict the observations (Asmi et al., 2012; Che et al., 2017; Hu et al., 2020; Zhang et al., 2017). Further investigation on how the changes in aerosol composition and hygroscopicity affect CCN variability at these sites is required.

In this study, we investigate OA sources, their temporal variability and their influence on CCN predictions at a high-altitude mountain site during an intensive summer field campaign. To understand the influence of aerosol composition on CCN, we calculate the overall aerosol hygroscopicity from bulk chemical composition measurements and then assume different OA



schemes to retrieve $\kappa_{OA}$. We focus the analysis on the influence that OA might have on the CCN predictions under different atmospheric conditions throughout the day. Additionally, a non-analytical model approach using neural networks was developed to predict CCN concentrations based on ancillary information on particle number concentration, OA mass fraction and oxygenation degree, and surface solar radiation.

## 2 Measurements

### 2.1 Experimental site

Aerosol measurements presented in this study were conducted at Sierra Nevada station (SNS) from 8 June to 13 July 2021 in the frame of the BioCloud field campaign (Jaén et al., 2023). The main objective of the campaign was to evaluate the impact of biogenic and anthropogenic emissions on the CCN budget at this high-altitude mountain site. SNS is located in the Sierra Nevada Mountain range in south-eastern Spain (37.10°N, 3.39°W, 2500 m a.s.l.), which is part of AGORA (Andalusian Global

Observatory of the Atmosphere). Measurements at SNS are performed following ACTRIS (Aerosol, Cloud and Trace gases Research Infrastructure, http://actris.eu) standards for in-situ measurements at high-altitude observatories (Pandolfi et al., 2018) and the station is part of the NOAA Federated Aerosol Network (NFAN, Andrews et al., 2019).

SNS is located at a horizontal distance of 21 km and an altitude difference of 1820 meters from the city of Granada which is located downslope of the mountains in a valley. Granada is a medium-size city with a population of 232.208 (www.ine.es,

2018), which increases up to 530.000 if the wider metropolitan area is considered. The main local aerosol source in Granada is road traffic, including both motor vehicle exhaust and re-suspension of particulate material from the roadways (Casquero-Vera et al., 2021; Rejano et al., 2023; Titos et al., 2014) . These pollutants emitted at Granada area can influence the aerosol properties observed in Sierra Nevada (Rejano et al., 2023). Atmospheric aerosol at SNS has been reported to be affected by the transport of particles from Granada metropolitan area because of planetary boundary layer (PBL) growth and the mountain-

valley breeze phenomenon (Rejano et al., 2021; Jaén et al., 2023; Casquero-Vera et al., 2020). Aerosol sources at SNS during summertime are primarily related to transport of pollutants from lower altitudes and regional transport, biogenic emissions from the vegetation, and desert dust transported from the Sahara Desert (Jaén et al., 2023). Furthermore, new particle formation (NPF) events are relatively frequent at midday, representing another important source of aerosol particles at this site (Casquero-Vera et al., 2020; Rejano et al., 2021; De Arruda Moreira et al., 2019).

### 2.2 Aerosol sampling and instrumentation

Sample air for all instruments was obtained through a stainless-steel tube located in the rooftop of the observatory, which is a three-story building. Inside this tube there are several smaller stainless-steel pipes, which provide sample air to the different instruments (Baron and Willeke, 2001). All measurements reported here refer to ambient conditions and were performed without aerosol size cut. Further information about the observatory and experimental conditions can be found in previous



studies performed at SNS (Casquero-Vera et al., 2020; Rejano et al., 2021; Jaén et al., 2023). In the following we describe the instruments used in this study.

A time-of-flight aerosol chemical speciation monitor, ToF-ACSM, (Fröhlich et al., 2013; Aerodyne Research Inc., Billerica, USA) was deployed to measure mass concentration and chemical composition of non-refractory submicron aerosol particles (NR-PM$_1$) with a 10-minute time resolution. The chemical species determined by the instrument are OA, SO$_4^{-2}$, NO$_3^-$, NH$_4^+$

and Cl$^-$. The instrument was operated at a flow rate of 3 lpm, and the air sample passed through a nafion dryer, maintaining the incoming relative humidity below 40%. During the campaign, the sample flow into the instrument was 0.108 lpm. A PM$_1$ standard aerodynamic lens focuses the sample flow into a narrow beam and transmits particles with vacuum aerodynamic diameter between 70 and 700 nm (Liu et al., 2007). Non-refractory particles are flash vaporized at 600 ºC with a tungsten vaporizer and ionized by electron impact at 70 eV. The instrument is equipped with a capture vaporizer that enhances

vaporization and gives a collection efficiency of 1. After sample ionization, the ions are introduced into a time-of-flight mass spectrometer (ETOF, Tofwerk Inc.) where they are orthogonally extracted and separated according to their mass-to-charge ratio (m/z). The mass spectra are obtained for m/z ions ranging from 12 to 200 Th. Finally, the mass spectral signals are converted to mass concentration (in µg/m3) using the ionization efficiency calculated from calibration curves of known reference species (Fröhlich et al., 2013).

Flow calibrations for the ToF-ACSM were performed before and after the BioCloud field campaign. The relative ionization efficiency (RIE) calibrations for NO$_3^-$ and SO$_4^{-2}$ were performed before the campaign using dry, size-selected 300 nm particles of ammonium nitrate and ammonium sulphate generated by an aerosol generator atomizer (TSI 3076). For more details about the ToF-ACSM calibrations see Fröhlich et al. (2013). Data processing was performed using the data analysis package "Tofware" (version 2.5.13, https://www.tofwerk.com/software/tofware/) running in the Igor Pro 7 environment (Wavemetrics

Inc., Oregon, USA). Data were corrected for changes during the campaign of the sample flow rate and N$_2$ signal (m/z 28), which is assumed to be constant in the atmosphere.

The CCN measurements were performed using a cloud condensation nuclei counter (CCNc) (Droplet Measurement Technologies, model CCN-200), which is based on a cylindrical continuous-flow thermal-gradient diffusion chamber where constant temperature gradients are applied, generating different SS conditions (Roberts and Nenes, 2005). One of the columns

sampled polydisperse particles, while the other column was connected to a differential mobility analyzer (DMA) to measure size-resolved CCN. For both columns, CCN concentrations were measured at four SS values: 0.2, 0.4, 0.6 and 0.8%, taking 10 minutes at each SS value. Only polydisperse measurements at 0.2, 0.4 and 0.6% SS are shown in this study. To ensure data quality due to instabilities of the instrument at each SS, CCN concentrations were filtered according to Rejano et al. (2021) criteria to ensure that NCCN measured at SS that differed by more than 20% from the SS set-point were disregarded. The total

flow rate of the instrument was fixed at 0.5 lpm with an aerosol flow of 0.05 lpm and sheath flow of 0.45 lpm. The flow rates were calibrated onsite before and after the campaign and checked regularly during the campaign. SS calibration using monodisperse ammonium sulfate was also performed onsite at the beginning and at the end of the campaign following the




procedure described in ACTRIS guidelines (http://actris.nilu.no/Content/SOP). Both calibrations provided satisfactory results and showed no change in instrument performance.

The particle number size distribution (PNSD) was measured in the mobility diameter range between 12-535 nm every 5 minutes using a scanning mobility particle sizer (SMPS, TSI model 3938), composed of a differential mobility analyzer (DMA, TSI 3081) and a condensation particle counter (CPC; TSI 3750). The aerosol flow rate was 1 lpm and the sheath flow was 5 lpm. The quality of the SMPS measurements were assured by frequently checking the flow rates and performing 203 nm PSL checks, following the ACTRIS and Global Atmospheric Watch (GAW) recommendations (Wiedensohler et al., 2012).

An aethalometer (Model AE-33, Magee Scientific) was used to determine the equivalent black carbon (eBC) mass concentration with a time resolution of 1 min. The aethalometer draws the ambient air at a constant flow rate of 4 lpm. The eBC is determined from the aerosol absorption coefficient at 880 nm using a mass absorption cross section of 7.77 $m^2$/g as recommended by the manufacturer. The $PM_1$ mass concentration was estimated as the sum of the mass of non-refractory components obtained by ToF-ACSM and eBC mass concentration measured by the aethalometer as suggested by the second

deliverable of Cost Action CA 16109 Colossal.

Finally, a Hukseflux LP02-05 pyranometer was used to measure the surface global solar radiation with 5-minute resolution.

# 3 Methodology

## 3.1 Source apportionment of organic aerosol.

The source apportionment of organic aerosol was performed using the positive matrix factorization (PMF) method (Paatero

and Tapper, 1994) using the multilinear engine ME-2 (Paatero, 1999). The PMF is a multivariate factor analysis technique that allows the decomposition of the measured OA mass spectral matrix (X), where the matrix columns are the variables (m/z ions) and the matrix rows are the observations (ToF-ACSM timestamps), into two matrices: the factors or sources profiles matrix (F) and the contributions matrix (G):

$$x_{ij} = \sum_{k=1}^{p} g_{ik} \cdot f_{kj} + e_{ij} \quad (1)$$

where $e_{ij}$ represent the elements of the residual matrix (E), accounting for unexplained information of X in the p factors solution. The number of PMF factors, p, is a pre-set parameter that must be established. Once the number of factors is fixed the algorithm solves Equation 1 iteratively, minimizing the Q function which is defined as:

$$Q = \sum_{i,j} \left(\frac{e_{ij}}{\sigma_{ij}}\right)^2 \quad (2)$$

where $\sigma_{ij}$ are the measurement uncertainties corresponding to the $x_{ij}$ input data. The solution with the correct number of

factors should give $Q/Q_{exp}$ near unity, with $Q_{exp}$ being the expected value of Q and is calculated as $Q_{exp} = n \cdot m - p \cdot (n+m)$; being n the number of observations and m the number of variables).



To improve the source apportionment characterization and achieve environmentally meaningful solutions, the ME-2 methodology allows establishment of a priori mass profiles of known OA sources, the so-called anchor profiles, based on previous scientific knowledge at experimental site (Canonaco et al., 2013) or on chamber data. The strength of this a priori constraint is modulated through the a-value approach (Paatero and Hopke, 2009; Brown et al., 2012). The a-value establishes how much deviation from the anchor profile the model allows to the solution factor. Thus, a fully constrained factor has an a-value=0, whereas for unconstrained factors the a-value=1. The ME-2 engine initialization and the results analysis was done using the SoFi v.8 toolkit (Source Finder, Canonaco et al., 2013) for Igor Pro environment. The PMF was run for a range of solutions from 3 to 5 factors and the mass spectra considered ranged between 12 and 120 Th, since higher m/z ions contribute only marginally to the mass spectra and exhibit low signal-to-noise ratio (SNR<0.2).

**3.2 CCN estimations and activation properties using κ-Köhler theory.**

The $\kappa$-Köhler theory establishes a mathematical relation between water vapor supersaturation ratio, critical diameter (which is the threshold size at which particles become CCN, $D_{crit}$) and $\kappa$ parameter (Petters and Kreidenweis, 2007). Therefore, from the overall $\kappa$ of an aerosol population we can estimate the $D_{crit}$ at a certain SS using $\kappa$-Köhler theory. This method assumes a homogenous aerosol population mixture (internally mixed) where all particles larger than this cutoff diameter activate (Jurányi et al., 2011). Thus, CCN number concentration, $N_{CCN}$, is estimated summing up the PNSD from Dcrit to the upper limit of the size distribution as follows:

$$N_{CCN}(SS) = \int_{D_{crit}(SS)}^{D_{max}} \frac{dN}{d \log D} d \log D \quad (4)$$

Alternatively, we can do the inverse calculation integrating the PNSD from its upper limit to the diameter at which the integral value equals the simultaneously measured $N_{CCN}$(SS) with the CCNc. Then, the effective hygroscopicity parameter can be retrieved using $\kappa$-Köhler theory from aerosol size distribution and CCN concentration measurements (Jurányi et al., 2011). These CCN-derived $\kappa$ values ($\kappa_{CCN}$) quantify the effective hygroscopicity of activated particles in the CCNc and exhibit a dependency on SS (Kammermann et al., 2010).

**3.3 Estimation of aerosol hygroscopicity from chemical composition measurements.**

One of the most commonly used approaches to estimate the total aerosol hygroscopicity from chemical composition measurements is based on the Zdanovskii-Stokes-Robinson (ZSR) approach. Considering ambient aerosols as a mixture of individual compounds, the hygroscopicity parameter can be retrieved using a mixing rule in terms of the volume fractions of the chemical species (Petters and Kreidenweis, 2007) as follows:

$$\kappa_{chem} = \sum_i \kappa_i \varepsilon_i \quad (3)$$

where $\varepsilon_i$ is the volume fraction of each chemical species and $\kappa_i$ its corresponding hygroscopicity. This approximation provides a successful explanation of observations as shown in previous studies (Bougiatioti et al., 2009; Rose et al., 2010; Wang et al., 2010; Bougiatioti et al., 2016). The summation is performed over all chemical species considered for the calculation of $\kappa_{chem}$



parameter. In this study we have considered three main terms in equation 3: OA, inorganic aerosol (IA) and eBC. Thus, $\kappa_{chem}$ can be estimated as follows:

$\kappa_{chem} = \kappa_{OA}\varepsilon_{OA} + \kappa_{IA}\varepsilon_{IA} + \kappa_{BC}\varepsilon_{BC}$      (4)

where $\kappa_{OA}$ ($\varepsilon_{OA}$), $\kappa_{IA}$ ($\varepsilon_{IA}$) and $\kappa_{BC}$ ($\varepsilon_{BC}$) are the hygroscopicity parameters (volume fractions) of organic aerosols, inorganic aerosols, and BC, respectively. The contribution of IA to $\kappa_{chem}$ considers some inorganic salts (ammonium nitrate, ammonium sulfate, ammonium bisulfate and sulfuric acid) present in the atmosphere. The volume fractions of these inorganic salts were obtained by the simplified ion pairing scheme presented by Gysel et al. (2007) using the inorganic species measured by the

ToF-ACSM ($SO_4^{-2}$, $NO_3^-$ and $NH_4^+$ ions). The density and hygroscopicity parameter for each inorganic salt were taken from previous studies (Kuang et al., 2020b; Wu et al., 2016) and are summarized in Table S1.

The inorganic contribution to $\kappa_{chem}$ is assumed to be a well-defined term in Equation 4. We assumed that BC particles are completely hydrophobic ($\kappa_{BC}$=0) for calculating $\kappa_{chem}$, which is a reasonable assumption as suggested in previous studies (Deng et al., 2019; Kuang et al., 2020b; Schmale et al., 2018). Unlike inorganic species that exhibit a well characterized

hygroscopic behavior, the water uptake capacity of OA species is poorly understood because of the presence of diverse organic species (Casans et al., 2023; Hallquist et al., 2009; Kanakidou et al., 2005; Rastak et al., 2017). This diversity makes determining $\kappa_{OA}$ extremely challenging (Kuang et al., 2020a). In Section 4.2, we will present different OA schemes in term of the PMF solution to estimate $\kappa_{OA}$ assuming different density and hygroscopicity values for each OA source.

### 3.4 Performing non-analytical solutions for CCN predictions: neural networks.

Apart from analytical solutions based on predefined relationships between variables, non-analytical solutions like machine learning techniques have become a powerful alternative to predict certain variables using ancillary information as input. Indeed, neural networks has been applied with remarkable success in recent years for regression problems in the framework of atmospheric sciences (Biancofiore et al., 2017; Comrie, 1997; Spellman, 1999), to relate atmospheric variables with non-linear and highly complex behavior.

For our regression problem, we built a neural network which uses 4 input parameters and has $N_{CCN}$ as the output parameter. Our neural network consists of a two-layer feed-forward network with sigmoid hidden neurons and linear output neurons. We chose 10 neurons hidden layer after verifying that results didn't improve with more neurons. We used the back-propagation algorithm (Rumelhart et al., 1986) with Bayesian regularization (Foresee and Hagan, 1997) for training the network. Data were split in training, validation and test using 55% of the data for training, 20% of the data to halt training when generalization

stops improving (neural network validation), and the remaining 25% of data was used for testing. Each subset of data was obtained by randomly selection of observations. The entire modeling process was performed using the neural net fitting tool of MATLAB software.



## 4 Results

### 4.1 BioCloud field campaign overview.

In this section we present an overview of the $PM_1$ chemical composition including identification of OA sources  and the analysis of CCN activation properties from 8 June to 13 July 2021 within the framework of BioCloud field campaign.

#### 4.1.1 Sub-micron aerosol chemical composition and source apportionment.

The average $PM_1$ concentration during the campaign was 3.85±2.88 µg/m³, with 10-min average concentrations ranging from 0.15 to 15.3 µg/m³. Figure 1a shows the mean $PM_1$ concentration and relative contribution of the considered species (OA, $SO_4^{-2}$,

$NO_3^-$, $NH_4^+$, $Cl^-$, eBC) to the total $PM_1$. On average, the most abundant aerosol component is OA (2.68 µg/m³), followed by $SO_4^{-2}$ (0.46 µg/m³) and eBC (0.33 µg/m3), with relative contributions of 70%, 12% and 9%, respectively. Inorganic components ($SO_4^{-2}$, $NO_3^-$, $NH_4^+$, $Cl^-$) represent 20% of the total $PM_1$ concentration on average, indicating the large contribution of organics at this high-altitude remote site during summertime. Similar OA dominance is observed in remote sites worldwide in summertime (Fröhlich et al., 2015; Heikkinen et al., 2020; Jimenez et al., 2009; Ripoll et al., 2015; Zhang et al., 2007). The

most abundant inorganic component is $SO_4^{-2}$ due to the higher $SO_2$ oxidation rates under high insolation conditions that favor the formation of this compound (Pey et al., 2009; Titos et al., 2014). $NO_3^-$ and $NH_4^+$ species exhibit similar low mass concentrations (0.15 µg/m³) probably due to the high summer temperatures that favor the instability of ammonium nitrate. $Cl^-$ shows a negligible concentration, near to the detection limit of the instrument. The mean eBC mass concentration (0.33 µg/m³) is in the range of those previously observed at SNS (Rejano et al., 2021), and in the range of values reported at other high-

altitude remote sites during summer, with values ranging between 0.2 and 0.5 µg/m³ across all sites (Ripoll et al., 2015; Zeb et al., 2020; Gramsch et al., 2020).

To gain insight into the local and regional aerosol sources and the underlying atmospheric aerosol processes that control aerosol evolution, diurnal variations of the mass concentration of the measured aerosol species were investigated (Figure 1b). The mass concentration of inorganic species exhibited an increase throughout the day starting at 8:00 UTC (local time –2 h). $NH_4^+$,

$SO_4^{-2}$ and $NO_3^-$ mass concentrations followed a similar diurnal pattern. OA also increased at midday, but the increase is more sharped, reaching a maximum between 12:00-16:00 UTC. The eBC mass concentration increased more gradually, starting at 3:00 UTC and reaching a maximum at 11:00 UTC. Based on these diurnal patterns, inorganic species and eBC are most likely transported from the Granada urban area due to upslope mountain breezes and the increase of the PBL height during daytime. OA exhibits a larger increase in concentration at midday hours compared to the other species (Figure 1b), which might suggest

the influence of upslope transport but also to additional sources of OA in the vicinity of the measurement site (such as local emissions or secondary processes as nucleation).



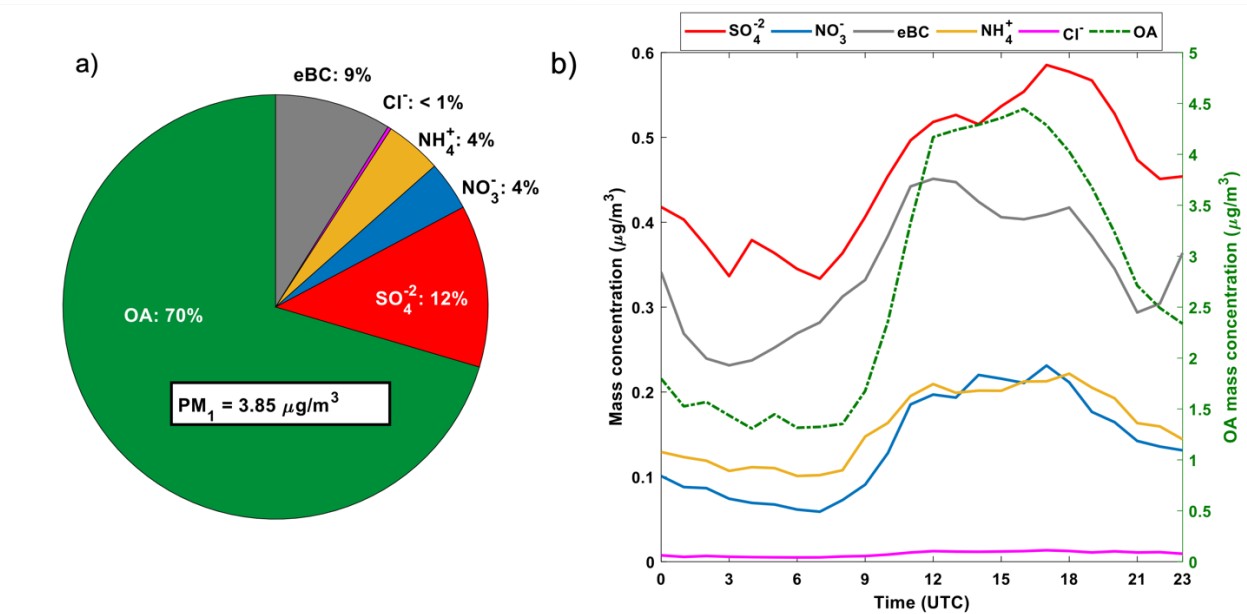

**Figure 1. a) Pie chart of PM1 inorganic species (SO4-2, NO3-, NH4+, Cl-), organic aerosol and eBC mass concentration averaged over the BioCloud campaign and b) mean diurnal pattern evolution for each species.**

Analyzing the OA mass spectra using PMF methodology, it is possible to infer whether the OA origin is locally formed and/or transported. To further explore the phenomenology of OA, the OA mass concentration was separated into different OA factors according to the PMF analysis. According to the $Q/Q_{exp}$ values and the physical interpretation of the PMF solution, the most reliable solution was the 3 factors solution with the following OA sources: hydrocarbon-like OA (HOA), less-oxidized oxygenated OA (LO-OOA) and more-oxidized oxygenated OA (MO-OOA). Once the three OA sources were identified, a new

constrained PMF solution was obtained to improve the source apportionment. We constrained the HOA factor to the Crippa et al. (2013) anchor profile, which is considered the standard mass profile for HOA, with an a-value=0.1. The LO-OOA and MO-OOA factors were kept unconstrained to adapt better to the site-specific aerosol characteristics.

The mass spectra profiles and the time-series for each OA factor are presented in Figure 2. As mentioned above, the first factor was constrained to the standard HOA profile and, therefore, the obtained mass spectrum has high contribution of $C_xH_y^+$

fragments (m/z 41, 43, 55, 57, 69, 71; Figure 2a1), also known as aliphatic hydrocarbons. These ions are typically related to primary emissions of diesel exhaust (Canagaratna et al., 2010; Crippa et al., 2013). The other two factors accounted for virtually all OA at SNS (around 95%) and were resolved freely by the model. Both secondary factors (LO-OOA and MO-OOA) are quite oxidized with large contributions of m/z 28 and 44 (Figure 2b1 and 2c1). OA at this site is mostly composed of oxygenated OA, which agrees with previous observations at mountain sites during summertime conditions when SOA

formation through photochemical oxidation is more efficient (Ripoll et al., 2015). The fraction of m/z 43 ($C_2H_3O^+$) and 44 ($CO_2^+$) ions relative to the whole mass spectra ($f_{43}$ and $f_{44}$, respectively) indicates the aerosol oxidation degree and allows





differentiation of the OOA into less oxidized OOA (i.e., LO-OOA with higher $f_{43}/f_{44}$ ratio) and more oxidized OOA (i.e., MO-OOA, lower $f_{43}/f_{44}$ ratio) (Fröhlich et al., 2015; Ng et al., 2010).

The results of the PMF show average contributions of 5%, 36% and 59% of HOA, LO-OOA and MO-OOA, respectively, to

the total OA concentrations during the measurement campaign. The low contribution of the HOA factor (which represents a 3.5% of the total $PM_1$ during the campaign), highlights the absence of important primary OA (POA) local sources close to the measurement site. However, sporadic peaks were observed throughout the field campaign (Figure 2a2), probably related to occasional local combustion emissions (Jaén et al., 2023). The first half of the campaign (before 26 of June) was characterized by a higher contribution of MO-OOA (mean values for this period were 2.0±1.4 and 0.7±0.8 µg/m³ for MO-OOA and LO-

OOA respectively), while LO-OOA becomes more relevant during the second half of the campaign (mean values for this period were 1.1±0.9 and 1.2±1.6 µg/m³ for MO-OOA and LO-OOA respectively) (Figure 2). The higher abundance of MO-OOA in the first half of the campaign might be associated with less efficient transport and predominance of stagnant conditions favoring the presence of aged aerosols, while the higher LO-OOA concentration might be associated with more efficient transport to SNS due to vertical transport of particles and gaseous precursors from lower altitudes by orographic buoyant

upward flows. These differences in the OA origin during each period can be related to different meteorological conditions for the two periods.

Figure S1 shows the timeseries of meteorological variables (temperature, pressure and relative humidity) during the campaign. The second half of the campaign is characterized by higher temperatures, higher pressure and lower relative humidity compared to the first half. These characteristics are likely to promote a more efficient mountain-valley breeze during the second half of

the campaign. To check the effectiveness of aerosol transport due to mountain-valley breeze regime, eBC concentration can be used as a tracer of transported aerosols from lower altitudes, due to absence of local BC sources. During the second half of the campaign, eBC shows a more pronounced diurnal pattern reaching higher concentrations during midday hours compared with the first half of the campaign (Figure S2). Thus, the atmospheric condition differences during the campaign could explain the predominance of each OA factor.




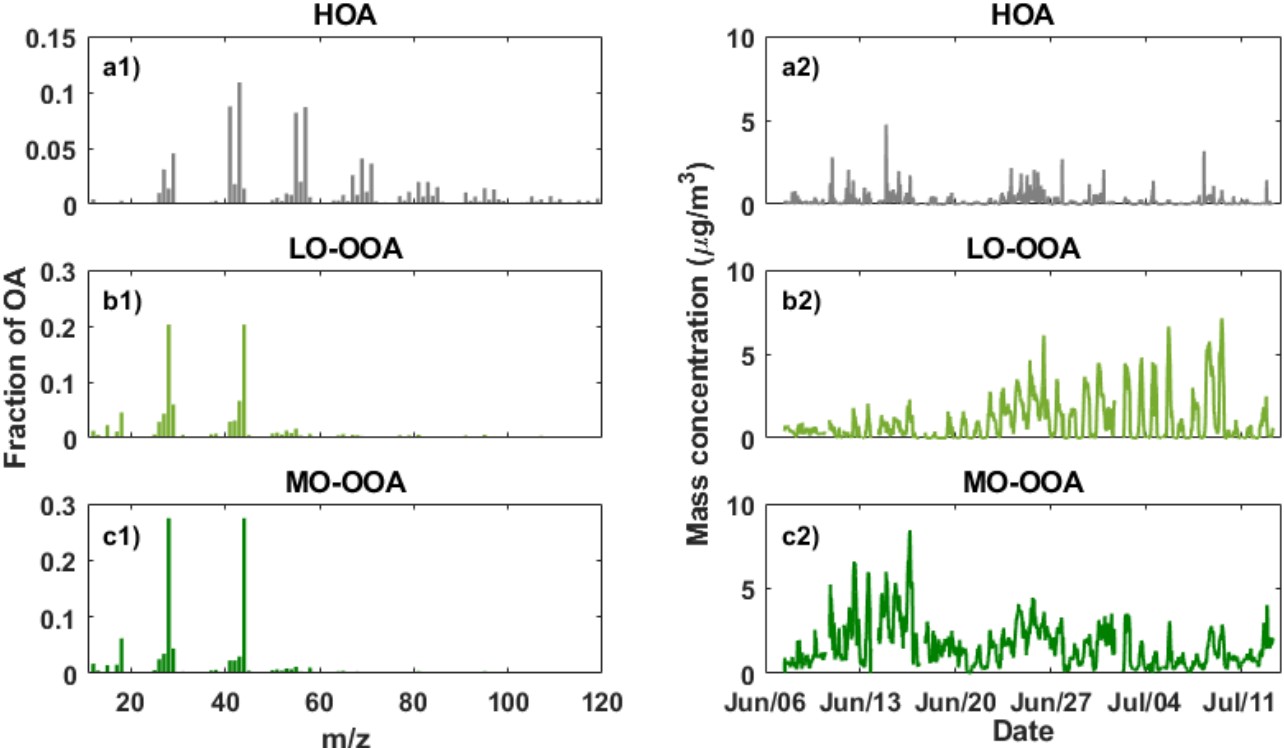

**Figure 2. Mass spectra of the three OA factors (left panels) and their time-series evolution (right panels) during the BioCloud field campaign.**

### 4.1.2 CCN activation properties

Aerosol chemical composition plays an important role in defining aerosol hygroscopicity and CCN activation properties (Svenningsson et al., 2006; Liu et al., 2018). In this sub-section we link some aerosol physical properties which are directly related to the CCN activity, such as total particle concentration ($N_{tot}$), nucleation mode particle concentration (defined as the concentration of particles below 25 nm, $N_{nucl}$), Aitken mode particle concentration (diameters between 25 and 100 nm, $N_{Ait}$), accumulation mode particle concentration (defined as the particle concentration above 100 nm, $N_{acc}$), and some activation parameters ($N_{CCN}$, $D_{crit}$, $\kappa_{CCN}$) at different SS values with the sub-micron chemical composition described previously. A statistical overview of these parameters (mean, median, standard deviation, percentiles 25 and 75) are shown in Table S2. The mean $N_{CCN}$ values ranged from 320±280 cm⁻³ at SS=0.2% to 800±700 cm⁻³ at SS=0.6%. The mean $D_{crit}$ value at SS=0.2% is 111±21 nm, indicating that particle activation is limited to accumulation mode particles. At higher SS some Aitken mode particles start to contribute to $N_{CCN}$, since mean $D_{crit}$ values decrease with SS (72±18 nm at 0.4% and 58±16 nm at 0.6%). In contrast, CCN-derived $\kappa$ values ($\kappa_{CCN}$) are mainly constrained to the range between 0.1 - 0.25 (which is the interquartile range for all SS, see Table S2), showing little dependence on SS with median values of 0.18, 0.15 and 0.13 at SS=0.2%, 0.4% and 0.6%, respectively. Overall, the aerosol activation properties agree with previous observations of these parameters at SNS





(Rejano et al., 2021; Casquero-Vera et al., 2020)  and with those reported at other mountain sites during summer season (Asmi et al., 2012; Georgakaki et al., 2021; Juranyi et al., 2011; Rejano et al., 2021).

To evaluate the influence of chemical species on the activation properties, Figure 3 shows the mean diurnal patterns of OA

factors, IA, eBC and particle concentration for each aerosol mode ($N_{nuc}$, $N_{Ait}$ and $N_{acc}$), along with $\kappa_{CCN}$ and particle and CCN number concentrations. All variables exhibit a clear diurnal pattern, but with some differences among them. Regarding particle number concentration in the different modes, $N_{nucl}$ exhibits a clear and sharp peak around midday hours (maximum at 14:00 UTC) due to the impact of new particle formation (NPF) events (Figure 3a). $N_{acc}$ exhibits a flatter pattern, with the increase in concentrations observed at midday mostly associated with vertical transport due to the mountain-valley breeze regime and

PBL height increase along the day. $N_{CCN}$ at all SS values follows a similar diurnal evolution (Figure 3b) as $N_{acc}$ with maximum CCN concentrations observed during the midday hours and minimum concentrations during nighttime.

The overall hygroscopicity of the activated particles ($\kappa_{CCN}$) exhibits an inverse diurnal pattern to the other aerosol variables (Figure 3d), with a decrease during morning and midday hours coinciding with the $N_{CCN}$ increase. This decrease of $\kappa_{CCN}$ is accompanied by an increase in the OA contribution to $PM_1$ (Figure 3d), however, it is not directly related because $\kappa_{CCN}$ starts

to decrease around 3:00 UTC and the OA/$PM_1$ ratio starts to increase around 6:00 UTC.  The OA/$PM_1$ ratio maximum (values higher than 0.75) was observed between 12:00-15:00 UTC due to the higher relative increase of LO-OOA and MO-OOA with respect to IA and eBC during those hours (Figure 3c) coinciding with the $\kappa_{CCN}$ minimum between 13:00-14:00 UTC for all SS. Figure 3c reveals that all species are affected by vertical upslope transport during morning and midday hours, however, LO-OOA can be also affected during midday hours by SOA formation linked to photochemical oxidation induced by high

concentration of $O_3$ and $NO_x$ (Figure S3a) together with high temperatures (Figure S3b) (Minguillón et al., 2016; Via et al., 2021). During nighttime, we observed the highest values of $\kappa_{CCN}$; this is probably related to the large contribution of inorganics to $PM_1$ in this period since IA species have the highest hygroscopicity values. At all SS investigated, $\kappa_{CCN}$ values are very similar during nighttime (around 0.32), while $\kappa_{CCN}$ differences among SS values are enhanced during midday hours (Figure 3d). This difference is likely due to the aerosol population becoming more dominated by OA (mainly LO-OOA) during midday

and requiring higher SS to activate less hygroscopic particles. Note that the diurnal pattern of $\kappa_{CCN}$ at all SS is constrained between 0.15 to 0.3, which is in the typical range for hygroscopic organic species (Kuang et al., 2020a), in agreement with the predominance of MO-OOA in our $PM_1$ measurements. These observations indicate that OA and its oxygenation degree (higher or lower contribution of MO-OOA/LO-OOA) might be an important factor controlling the overall aerosol hygroscopicity at SNS during the day.



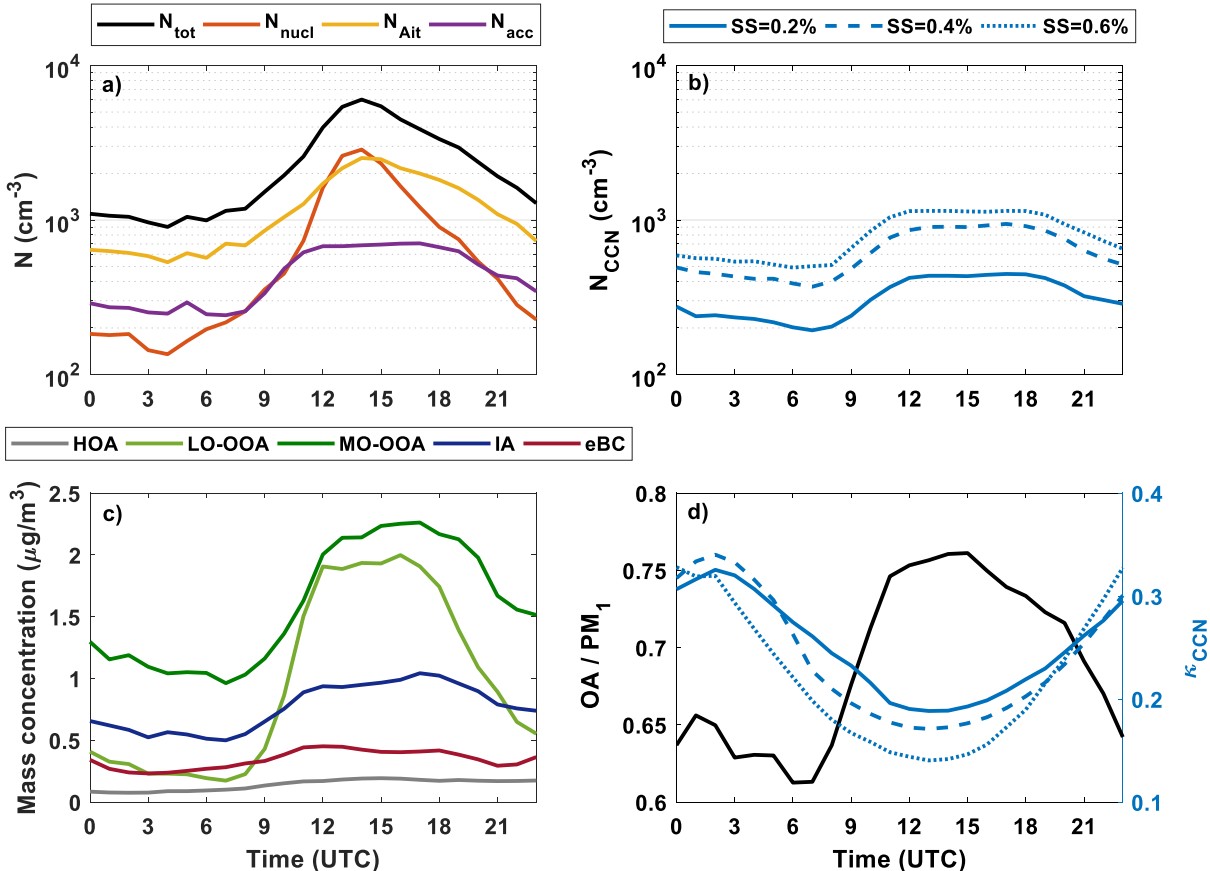

**Figure 3. Mean diurnal pattern of a) particle number concentration and each aerosol mode (N$_{tot}$, N$_{nucl}$, N$_{Ait}$, N$_{acc}$); b) CCN concentrations, c) OA factors and IA species mass concentration and d) OA and PM$_1$ ratio and CCN-derived kappa.**

### 4.2 Predicting CCN concentration: role of organic aerosol.

In the previous section we observed that the diurnal variability of the κ parameter might be related with the OA content as well as with its oxidation degree. In this section we apply different approaches to predict CCN concentrations and evaluate the impact of OA sources in the overall performance of the closure scheme depending on the underlying assumptions. We use the total aerosol hygroscopicity calculated from PM$_1$ chemical composition measurements ($\kappa_{chem}$) using three different organic hygroscopicity schemes for CCN calculation and discuss the degree of agreement of the different CCN closures under different atmospheric conditions. Then, another approach to estimate $\kappa_{OA}$ in terms of the f$_{44}$ parameter is presented to link the hygroscopicity changes with the aerosol oxidation degree. Finally, a neural network-based approach using ancillary parameters is used to predict the CCN concentrations at Sierra Nevada.



### 4.2.1 Using different OA hygroscopicity schemes

Using the bulk chemical composition measurements, we estimated the overall $\kappa_{chem}$ as explained in Section 2.4 to predict $N_{CCN}$ using $\kappa$-Köhler theory and PNSD data. For the IA contribution to $\kappa_{chem}$, the Cl⁻ species was neglected due to its low
contribution at SNS (Cl⁻ concentrations were very close to the detection limit of the instrument), as shown in Section 4.1.1. In this study we used different $\kappa$ values for the obtained OA factors (HOA, LO-OOA and MO-OOA) to compute the overall $\kappa_{chem}$ in three different ways:

- Scheme 1: we assume that $\kappa_{HOA} = \kappa_{LO-OOA} = \kappa_{MO-OOA} = 0.1$, which is the typical value observed for $\kappa_{OA}$ in a wide variety of environments (Gunthe et al., 2009; Jurányi et al., 2011; Rose et al., 2010; Schmale et al., 2018).
- Scheme 2: we assume that HOA are hydrophobic particles, $\kappa_{HOA} = 0$ (Cappa et al., 2011; Jimenez et al., 2009; Kanakidou et al., 2005; Thalman et al., 2017), and LO-OOA and MO-OOA components have a constant $\kappa$ value of 0.1.
- Scheme 3: since the level of oxidation of OA affects its hygroscopicity, we assume specific hygroscopicity values for LO-OOA and MO-OOA ($\kappa_{LO-OOA} = 0.08$ and $\kappa_{MO-OOA} = 0.16$) as reported by (Cerully et al., 2015). HOA is again
assumed to be non-hygroscopic ($\kappa_{HOA} = 0$).

Table 1 summarizes the densities and hygroscopicity values of HOA, LO-OOA and MO-OOA used for calculating the $\kappa_{chem}$ value for the different OA schemes. The volume fractions of OA components were obtained assuming the density of OOA as 1.4 g/cm³ and for HOA the typical POA density of 1 g/cm³ was assumed (Kuang et al., 2020a; Wu et al., 2016).

**Table 1. Assumed densities and hygroscopicity values for each OA factor in the different OA schemes.**

| *OA factor* | *Parameter* | | | |
|---|---|---|---|---|
| | $\rho$ (g/cm³) | $\kappa$ | | |
| | | Scheme 1 | Scheme 2 | Scheme 3 |
| **HOA** | 1 | 0.1 | 0 | 0 |
| **LO-OOA** | 1.4 | 0.1 | 0.1 | 0.08 |
| **MO-OOA** | 1.4 | 0.1 | 0.1 | 0.16 |

Figure 4 shows the violin plots of the retrieved $\kappa$ values for each OA scheme, $\kappa_{chem}$, and the calculated $\kappa$ values from the CCNc measurements, $\kappa_{CCN}$, at different SS. The $\kappa_{chem}$ values exhibit lower variability (ranging from 0.1 to 0.35) compared to the $\kappa_{CCN}$ values (from 0.06 to 0.7). The probability density function (PDF) of $\kappa_{chem}$ for schemes 1 and 2 are very similar, with
maximum around 0.14. The main difference in the data distribution between both schemes is observed at low hygroscopicity



values, which have been identified as periods of higher HOA contribution (i.e., during HOA peak events). Scheme 3 exhibits a clearly different data distribution compared to schemes 1 and 2 due to the assumption of a time dependent $\kappa_{OA}$ in terms of the relative contribution of LO-OOA and MO-OOA factors. In general, scheme 3 results in higher $\kappa_{chem}$ values (mean and median values are 0.20) since we assumed a higher hygroscopicity for the MO-OOA factor, which is the main factor controlling

OA at SNS. Also, Figure 4 shows that the data is more homogenously distributed around the mean value for scheme 3, while the distributions for schemes 1 and 2 are skewed towards lower values.

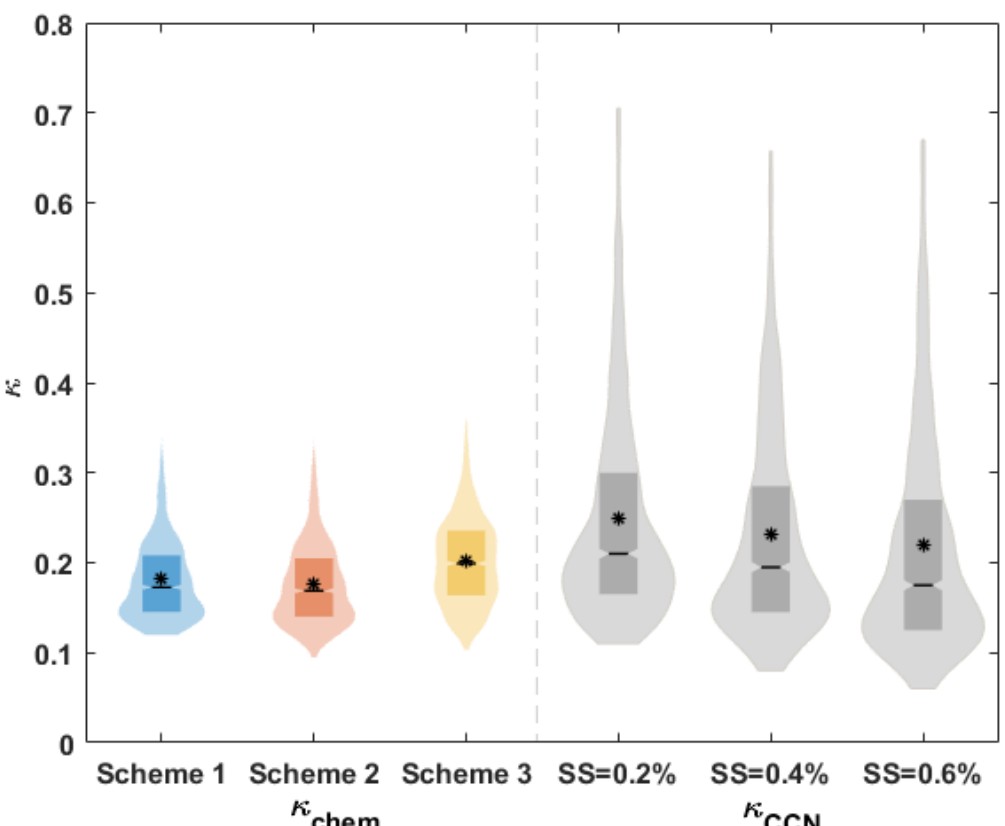

**Figure 4. Violin plot of κ distribution data for the chemical schemes (κ$_{chem}$) and the CCN calculation at different SS values (κ$_{CCN}$). The boxes represent the interquartile distance, and the asterisk is the mean value.**

The $\kappa_{CCN}$ values exhibit very different data distributions relative to the $\kappa_{chem}$ values. All $\kappa_{CCN}$ PDFs show a clear maximum and positive skewness with some outlier observations (higher mean than median values). This is likely due to the larger variability of the parameters used to retrieve aerosol hygroscopicity in the supersaturated regime (i.e., N$_{CCN}$ and PNSD via $\kappa$-Köhler theory) compared to smaller changes in chemical composition. This is particularly important in the case of SNS since the sub-micron chemical composition is dominated by OA, and despite the changes in hygroscopicity among OA constituents,

the range of change in $\kappa_{chem}$ is quite limited, because $\kappa_{chem}$ is less sensitive to temporal changes in composition. As




anticipated, higher SS values result in a shift to lower values in the data distribution due to activation of less hygroscopic particles. Across all SS values, mean $\kappa_{CCN}$ are higher than $\kappa_{chem}$ for the different OA schemes. However, the differences between $\kappa_{CCN}$ and $\kappa_{chem}$ median values are minimal. It is important to note that the $\kappa_{CCN}$ accounts only for activated particles in the CCNc, whereas $\kappa_{chem}$ accounts only for aerosol particles in the size range allowed by the aerodynamic lens in the ToF-

ACSM. Therefore, depending on the SS, both instruments may be measuring particles in different size ranges (as mentioned in Section 2.2). This effect might have the largest influence at SS=0.6% because the median $D_{crit}$ values is 60 nm which is below the optimum size range of the ToF-ACSM (70-800 nm). Moreover, both methods assume internally mixed particles to estimate the overall $\kappa$, which is an important limitation in the case of externally mixed particles (Wang et al., 2010; Ren et al., 2018; Kulkarni et al., 2023).

Based on the calculated $\kappa_{chem}$ values we estimated $N_{CCN}$ using $\kappa$-Köhler theory with a time-resolution of 30-minutes. In addition, we have used a simpler approach to estimate $N_{CCN}$ from PNSD data, which consists in assuming that particles above a certain size are activated. In this case, we selected 80 nm as the fixed activation diameter and $N_{80}$ (number concentration of particles with diameter larger than 80 nm) is used as a proxy for $N_{CCN}$. This threshold diameter has been selected because at medium SS values (0.4-0.5%), the $D_{crit}$ for a wide variety of aerosol types is constrained between 70-90 nm (Rejano et al.,

2023). The comparison between predicted and measured $N_{CCN}$ at the different SS values for the different OA schemes is shown in Figure 5. The results show that CCN closure dependence depends on SS when the $N_{80}$ approach is used. This is expected since this simple approach does not include the $D_{crit}$ dependence with SS. In this case, the predicted $N_{CCN}$ values overestimate the measurements at low SS and underestimate the measurements at high SS level. At SS=0.4% the mean $D_{crit}$ is 72±18 nm and, therefore, despite the diurnal and day-to-day variability in $D_{crit}$ which might hamper the predictions using $N_{80}$, the $N_{80}$





proxy explains very accurately the $N_{CCN}$ observations at this specific SS with the best correlation coefficient (r=0.94) and slope

of the regression (1.06).



**Figure 5. Log-log scatter plot of predicted CCN concentrations ($N_{CCN}$ pred) as a function of observed CCN concentrations ($N_{CCN}$ obs) using the four prediction schemes. The solid blue line represents the 1:1 line and the dashed lines are the +/-10%. The linear**
**equation and Pearson correlation coefficient (r) is also included.**

For the chemical CCN closure approach (OA schemes 1-3), all the schemes overestimate the CCN observations with slope

values ranging from 1.08 to 1.4 and correlation coefficients between 0.89-0.94 (Figure 5), indicating similar CCN closure for

all SS and schemes. A slightly worse agreement between predictions and observations is observed at SS=0.2% probably due



to higher discrepancy between $\kappa_{CCN}$ and $\kappa_{chem}$ at this SS, as previous studies have pointed out for low SS values (Cai et al.,
2018; Mei et al., 2013). Closure results for schemes 1 and 2 are very similar, despite the observed difference in $\kappa_{chem}$ values
between both schemes. This similarity is due to the low contribution of HOA at SNS. For scheme 3, the results indicate that
assuming a lower/higher $\kappa$ for LO-OOA/MO-OOA, respectively, rather than the standard $\kappa_{OA}$ =0.1, leads to a larger
overestimation of the predicted $N_{CCN}$ (especially at SS=0.2). Moreover, scheme 3 results show no improvement in the
correlation coefficients compared to the other OA schemes. Despite the large variability observed in the OA components, our
results demonstrate that the simple approach of assuming a constant $\kappa_{OA}$ of 0.1, even for a complex environment dominated
by OA, seems to provide satisfactory predictions of CCN concentration.

These results agree with other CCN closures studies based on bulk chemical composition under varying assumptions of OA
hygroscopicity (e.g., Kulkarni et al., 2023b; Meng et al., 2014; Ren et al., 2018b; Zhang et al., 2017b). Mei et al. (2013)
obtained good CCN closures at OA-dominated conditions (70-80% of PM$_1$) assuming a constant $\kappa_{OA}$ values of 0.08 and 0.13
(which are very close to $\kappa_{OA}$ = 0.1 used in this study). Rose et al. (2011) reported $N_{CCN}$ overestimations of 20% assuming $\kappa_{OA}$
= 0.1 near Guangzhou area (China), but better results (overestimation of 10%) were observed when further assumptions about
the hygroscopicity of low volatility particles were included. Assuming $\kappa_{OA}$ = 0.1 using both bulk and size-resolved chemical
composition, Meng et al. (2014) showed at a coastal site in Hong Kong that $N_{CCN}$ overestimations reached values of 26% and
10%, respectively. These authors concluded that CCN closures can be less sensitive to hygroscopicity considerations and some
mixing state considerations may play a role. In contrast, Ren et al. (2018b) demonstrated at an urban environment that aerosol
mixing state plays a minor role in CCN prediction when $\kappa_{OA}$ exceeds 0.1. They obtained good closure (closure ratios of 1.0-
1.16) using bulk chemical and internally mixture assumptions in the Beijing urban area under clean conditions. Siegel et al.
(2022) also obtained accurate $N_{CCN}$ closure results (slopes between 0.82-0.91) in the Arctic under internally mixed assumptions
by characterizing very precisely the organic hygroscopicity based on laboratory experiments and field observations. When
considering remote sites without the influence of local emissions, Cai et al. (2018) demonstrated that either bulk or size-
resolved chemical composition measurements can achieve practically the same agreement in $N_{CCN}$ predictions. Therefore, the
accuracy of $N_{CCN}$ predictions can exhibit a wide variety of results depending on the characteristics of the experimental site and
the atmospheric conditions.

To get a deeper understanding of the performance of CCN predictions and gain knowledge about how the differences in OA
composition during the day may or may not affect the CCN predictions, we calculated the diurnal evolution of the relative bias
([$N_{CCN}$ $^{pred}$-$N_{CCN}$ $^{obs}$]/$N_{CCN}$ $^{obs}$) of the $N_{80}$ approach and each OA scheme. Since the SS did not appear to cause significant
differences in the estimation of the CCN among the three OA schemes, from now on, we focus the analysis at SS=0.4%. Figure
6 shows the median diurnal evolution of the relative bias of each scheme for SS=0.4%. In this analysis we consider CCN
predictions to be accurate when the associated uncertainty is within the range of ±10% (grey shadowed area in Figure 6) which
is the instrument uncertainty associated with $N_{CCN}$ (Schmale et al., 2017). All schemes exhibit similar diurnal patterns in
relative bias with negative values during nighttime hours and positive during midday hours. There is a clear difference between
the relative bias pattern obtained by $N_{80}$ and the OA schemes. Figure 6a shows both the diurnal pattern of $D_{crit}$ at 0.4% and the





threshold size of 80 nm. As expected, the difference between the observed $D_{crit}$ and the assumed threshold size (80 nm) is clearly related to the bias value and, in general, the positive/negative bias is associated $D_{crit}$ values larger/smaller than 80 nm.

The largest deviations with respect to observations are found during nighttime hours (underestimation of $N_{CCN}^{obs}$ between 20-30%), when the $D_{crit}$ is considerably below 80 nm. Therefore, the use of this approach should be limited to situations when the $D_{crit}$ is fairly constant and restricted to a specific SS.

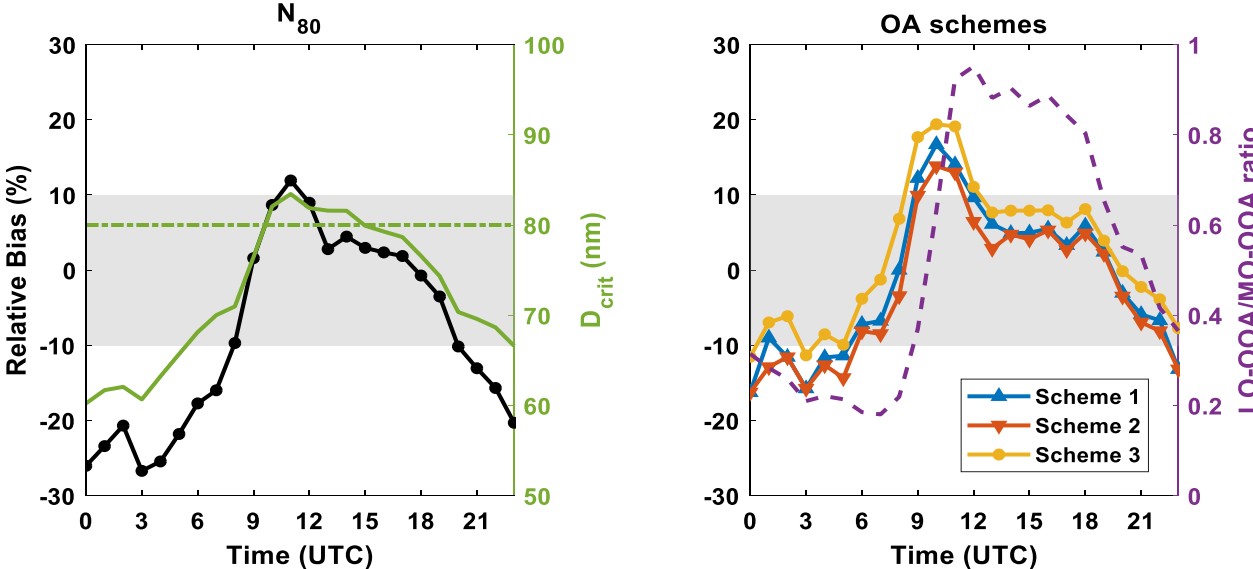

**Figure 6. Diurnal evolution of the median relative bias in CCN predictions at SS=0.4% for each prediction scheme. The grey shaded**
**area in all panels represents the ±10% relative bias. The $D_{crit}$ at SS=0.4% is shown in panel a) and the horizontal line represents the threshold size of 80 nm. The ratio between LO-OOA and MO-OOA mass concentrations shown in panel b).**

For the OA schemes, the diurnal evolution of the relative bias is similar to the $N_{80}$ approach with negative relative bias during nighttime and positive during daytime. The relative bias ranges from -6% to -16% for all OA schemes during the nighttime period which is a smaller range than observed for the $D_{crit}$ = 80 nm scheme. The nighttime period is associated with free
tropospheric conditions dominated by aged aerosol (OA is dominated by MO-OOA). Conversely, during morning/midday hours the relative bias increases from its minimum value at 6:00 UTC (3-8% underestimation) to its maximum value at 10:00 UTC (14-20% overestimation) (Figure 6b). The LO-OOA/MO-OOA ratio and relative bias diurnal patterns show similar shape (Figure 6b), but with 1 hour of delay between the maxima values for each parameter. This suggests that the largest bias occurs when the relative contribution of LO-OOA and MO-OOA starts changing. When the ratio LO-OOA/MO-OOA is constant the
relative bias remains constant as well. These results indicate that the relative bias in CCN predictions is highly dependent on the LO-OOA and MO-OOA variability and their relative contribution to OA. Since these factors have different degrees of oxidation, in the next section we present a new OA scheme that describes $\kappa_{OA}$ in terms of OA oxidation degree.




### 4.2.2 Parameterizing $\kappa_{OA}$ in terms of OA oxidation degree using f$_{44}$ parameter

In this sub-section we calculate $\kappa_{OA}$ based on $\kappa_{CCN}$ assuming that $\kappa_{IA}$ and $\kappa_{BC}$ are well-known terms in Equation 5 (Cerully et al., 2015; Kuang et al., 2020b; Thalman et al., 2017):

$\quad \kappa_{OA} = \frac{\kappa_{CCN} - \kappa_{IA}\varepsilon_{IA} - \kappa_{BC}\varepsilon_{BC}}{\varepsilon_{OA}}$ (5)

With this approach, the mean and median values of calculated $\kappa_{OA}$ are 0.18 and 0.15 at SS=0.4%, respectively, which are values higher than the standard value of $\kappa_{OA}$=0.1 but are within the range of ambient $\kappa_{OA}$ observations in the supersaturated regime (Levin et al., 2014; Gunthe et al., 2011; Kawana et al., 2016). The inferred values of $\kappa_{OA}$ confirm the predominance of MO-OOA species in the activated particles at SS=0.4% since it is very close to the assumed value of 0.16 for $\kappa_{MO-OOA}$ in the

OA scheme 3 (Table 1). Figure S3 shows the probability density function (PDF) for the effective $\kappa_{OA}$ at SS=0.4% retrieved with this method. The PDF distribution shows its maximum at $\kappa_{OA}$=0.11, which is similar to the assumed $\kappa_{OA}$ (~0.1) for scheme 1. The PDF also exhibits a clear positive skewness revealing the influence of more hygroscopic species at SS=0.4% Previous studies have parameterized $\kappa_{OA}$ as a function of the oxidation degree using the f$_{44}$ parameter (Kuang et al., 2020a). Therefore, we explore a potential improvement of the $\kappa_{OA}$ calculation at SNS by establishing a new OA scheme (named here

as scheme 4) based on a linear relationship between $\kappa_{OA}$ and f$_{44}$. This enables calculation of the $\kappa_{chem}$ as follows:

$\quad \kappa_{chem} = (m \cdot f_{44} + n)\varepsilon_{OA} + \kappa_{IA}\varepsilon_{IA} + \kappa_{BC}\varepsilon_{BC}$ (6)

where *m* and *n* are the slope and the intercept of the linear relationship between $\kappa_{OA}$ and f$_{44}$. To establish the parameterization, the dataset has been split randomly in two subsets: the first is used to obtain the linear regression and the second to check its performance for CCN calculation. Each data subset consists of 50% of the data. In the first subset of data, we re-sampled the

f$_{44}$ values into 80 bins and calculated the corresponding average $\kappa_{OA}$ values for each f$_{44}$ bin. Then, the empirical parameterization was obtained by establishing a linear regression between the averaged $\kappa_{OA}$ values and f$_{44}$. As shown in Figure 7 there is a clear linear trend between the binned values of $\kappa_{OA}$ and f$_{44}$. For high values of f$_{44}$ (especially above 0.26) the $\kappa_{OA}$ values exhibit higher dispersion. The $\kappa_{OA}$ and f$_{44}$ relationship obtained in this analysis (slope of 3.24) is for 0.2 < f$_{44}$ < 0.32. These high value of f$_{44}$ are due to the high oxidation degree of OA and the low contribution of HOA at this site. Previous

studies that reported a linear relationship between $\kappa_{OA}$ and f$_{44}$ were developed for less oxidized aerosol with f$_{44}$ values ranging from 0.05 to 0.20 (Duplissy et al., 2011; Kuang et al., 2020b; Chen et al., 2017; Mei et al., 2013). Those studies also reported lower slopes for the $\kappa_{OA}$-f$_{44}$ relationship, ranging between 2.1-2.4. Like the SNS analysis being reported on here, these studies from the literature were performed at OA-dominated sites during warm season. However, these other sites observed lower f$_{44}$ values due to a higher contribution of HOA and biomass burning related OA (Duplissy et al., 2011; Mei et al., 2013; Chang et

al., 2010). For fresh emitted biomass burning particles, Chen et al. (2017) also obtained a lower slope value of 2.3 associated with low f$_{44}$ values (f$_{44}$ <0.1). A significantly lower slope for the $\kappa_{OA}$-f$_{44}$ relationship (1.04) was reported by Kuang et al.



(2020a) for measurements on the north China plain during winter where the aerosol composition was dominated by higher contribution of HOA and coal combustion OA ($f_{44}$ <0.15 ). In contrast to these values reported in the literature, the $\kappa_{OA}$-$f_{44}$ relationship retrieved in our study is for higher values of $f_{44}$ and exhibits the largest slope. Our results therefore suggest that
the fit to the $\kappa_{OA}$-$f_{44}$ relationship depends on the oxygenation degree of the organic particles. However, some of the variability observed among studies in the $\kappa_{OA}$-$f_{44}$ relationship might arise from differences in $\kappa_{OA}$ calculation, such as the value of SS used and/or whether bulk or size-resolved measurements were available.

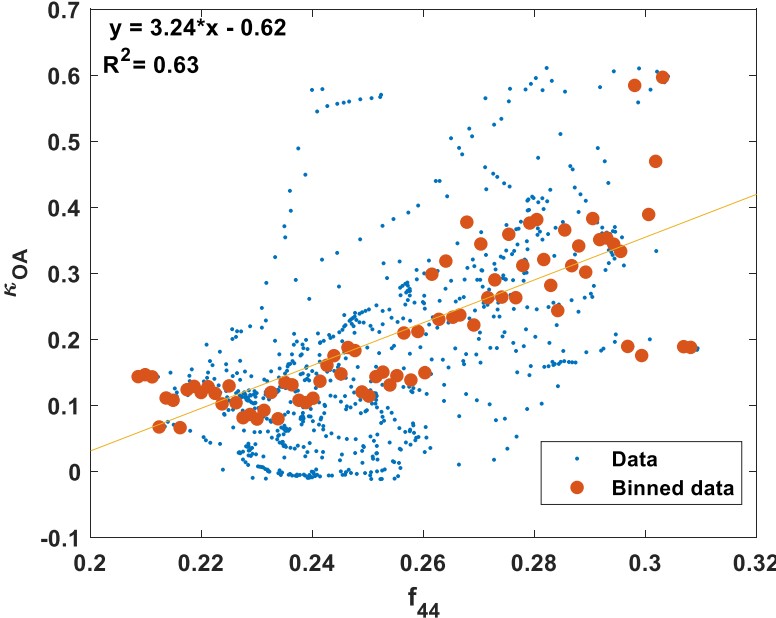

**Figure 7. Scatter plot of $\kappa_{OA}$ at SS=0.4% respect to $f_{44}$. The linear regression is applied to the binned data.**

After applying scheme 4 to the other half of the dataset, the CCN closure at SS=0.4% exhibits a similar slope and correlation coefficient (1.13 and 0.93, respectively, see Figure 8a) as the three other OA schemes. For high $f_{44}$ values (> 0.25) the CCN closure is better (slope of 1.05), while for low $f_{44}$ values (<0.25) the CCN predictions tend to overestimate the CCN concentrations and exhibit higher data dispersion. This is also observed in the median diurnal pattern of the relative bias (Figure 8b). During nighttime conditions (when high $f_{44}$ values are observed) the new OA scheme can explain the observations within
the ±10% range and improves the CCN closure relative to the previous OA schemes. However, the relative bias increases up to 35% (Figure 8b) during morning and midday hours when the aerosol is characterized by lower $f_{44}$ values associated with higher LO-OOA contribution. The sensitivity of $\kappa_{OA}$ to changes in $f_{44}$ is highly dependent on aerosol sources and atmospheric conditions, and significant deviations have been observed depending on the site (Kuang et al., 2020a). Our results are comparable with those of Zhang et al., (2016). They analyzed the impact of aerosol oxidation level on CCN predictions at a
suburban site in Northern China using the $\kappa_{OA}$-$f_{44}$ relation presented by (Mei et al., 2013). They showed that for OA mass fractions higher than 0.6 the $N_{CCN}$ predictions are very sensitive to $f_{44}$ values and the best CCN closures were observed for



$f_{44}$>0.15 with slope value around 0.94 at SS=0.39%. As observed in this study and in (Zhang et al., 2016), accurate $N_{CCN}$ predictions at OA-dominated sites using $\kappa_{OA}$-$f_{44}$ relation are challenging since both $\kappa_{OA}$ and $N_{CCN}$ are very sensitive to $f_{44}$ values.

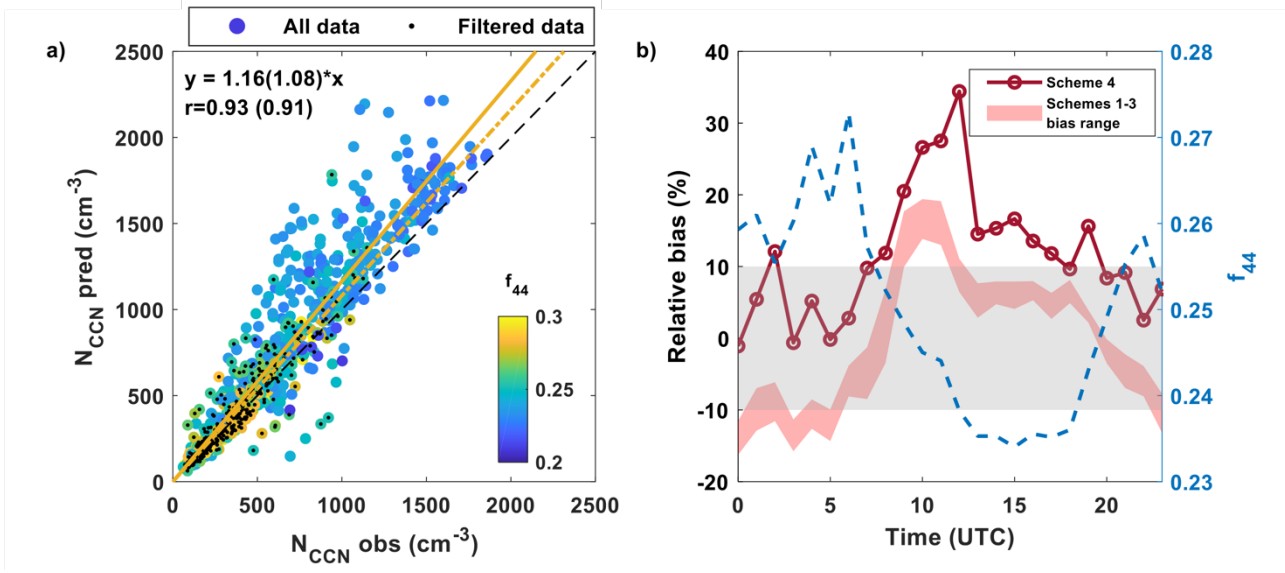

**Figure 8. a) Scatter plot of predicted CCN concentrations ($N_{CCN}$ pred) as a function of observed CCN concentrations ($N_{CCN}$ obs) using the OA scheme 4. Datapoints are colored-coded by the corresponding $f_{44}$ value. Datapoints with $f_{44}$>0.25 are also represented as small black dots The black dash line represents the 1:1 line. The linear equation and Pearson correlation coefficient (r) are also included for all data and for filtered data in parenthesis. The yellow solid and dash lines represent the linear regression of all and filtered data, respectively. b) Median diurnal evolution of the relative bias at SS=0.4% of the OA scheme 4 (left Y axis) and $f_{44}$ (right Y axis). The grey shaded area represents the ±10% relative bias. The red shaded area represents the relative bias range for the other OA schemes shown in Figure 6b.**

To sum up, the new $\kappa_{OA}$ calculation using $f_{44}$ parameterization shows good agreement between CCN calculations and observations during nighttime (bias ranging between 0-10% from 21:00 UTC to 6:00 UTC), however, it results in worse predictions during morning and midday hours at SNS. After verifying that all OA schemes for calculating $\kappa_{OA}$ yield nearly identical results, with the most significant biases occurring under conditions influenced by daytime vertical upslope transport of particles and/or NPF events, we conclude that using a bulk $\kappa_{chem}$ to predict $N_{CCN}$ consistently results in discrepancies with observations. The clear diurnal variability of aerosol properties and atmospheric conditions may require size-resolved chemical composition or mixing state assumptions for the aerosol population, like externally mixed aerosol or even combination of aerosol populations with different mixing state (Kulkarni et al., 2023; Zhang et al., 2017; Ren et al., 2018), to improve the results throughout the day.

### 4.2.3 Non-analytical approach for CCN prediction: neural networks

In this section, we develop a prediction scheme based on a neural network that uses 4 input parameters ($N_{80}$, OA/$PM_1$, $f_{44}$ and surface global radiation) to account for the main features affecting the CCN concentration at SNS, respectively: aerosol



concentration in the CCN-active size range, OA contribution to total $PM_1$ and its oxygenation degree (related to OA hygroscopicity) and insolation conditions that might affect secondary processes influenced by photochemistry. More details on the neural network architecture were explained in Section 3.4.

Figure 9 shows the performance of the neural network approach for CCN estimation at SS=0.4% (slope of 0.92 and correlation coefficient of 0.96). This neural network approach shows the best correlation with observations in this study. Comparing with

the analytical approaches using bulk chemical composition measurements, this model shows a general underestimation of measurements (slope <1) contrasting with the overestimation obtained for all OA scheme approaches (all slopes >1). In terms of capturing the diurnal variability, this new approach performs better, since the median diurnal pattern remains within the ±10% range (Figure 9b). The neural network can describe $N_{CCN}$ variability throughout the day, even during morning and midday hours when all four OA schemes exhibited the highest bias values. The neural network scheme also explains the CCN

variability during the most complex aerosol conditions at SNS. The inclusion of the surface global radiation in the neural network model acts as a proxy of photochemical activity and secondary processes influence which helps to diminish the overestimation peak observed for all OA schemes during the midday hours.

This newly developed model is able to manage the non-linear and time-dependent relationships between variables during the hours when aerosol population might be an external mixture of background particles, upslope transport particles and/or

particles produced during NPF events, suggesting it is a suitable approach for CCN prediction throughout the day. Park et al. (2023) also proposed machine learning approaches to develop CCN predictions based on multiple linear regression and non-negative matrix factorization techniques. They concluded that these methods are robust and capable of simulating either internal or external mixing conditions. However, the CCN predictions observed in our study are more accurate ($R^2$=0.88) than the results observed by Park et al. (2023) ($R^2$ between 0.71-0.81). This might be due to the input parameters considered in the

neural network, since Park et al. (2023) only considered aerosol size distribution measurements without any consideration of chemical composition or hygroscopicity. Nair et al. (2021) used a random forest regression model and also reported strong agreement between CCN estimations and observations during an aircraft campaign. They used model-simulated data of aerosol composition, atmospheric trace gases and meteorological variables without aerosol size information to estimate $N_{CCN}$ at SS=0.4%, finding a Kendall correlation coefficient of 0.76.

Our analysis along with these previous literature results indicate that machine learning approaches are very useful for accurately predicting $N_{CCN}$ under different conditions in terms of other aerosol properties. Further, this suggests that CCN coverage can be improved worldwide by using machine learning and making use of more routinely measured parameters such as aerosol size distribution, OA properties and global radiation. However, further studies assessing the potential of these tools at multiple sites and during long-time scales are still necessary.



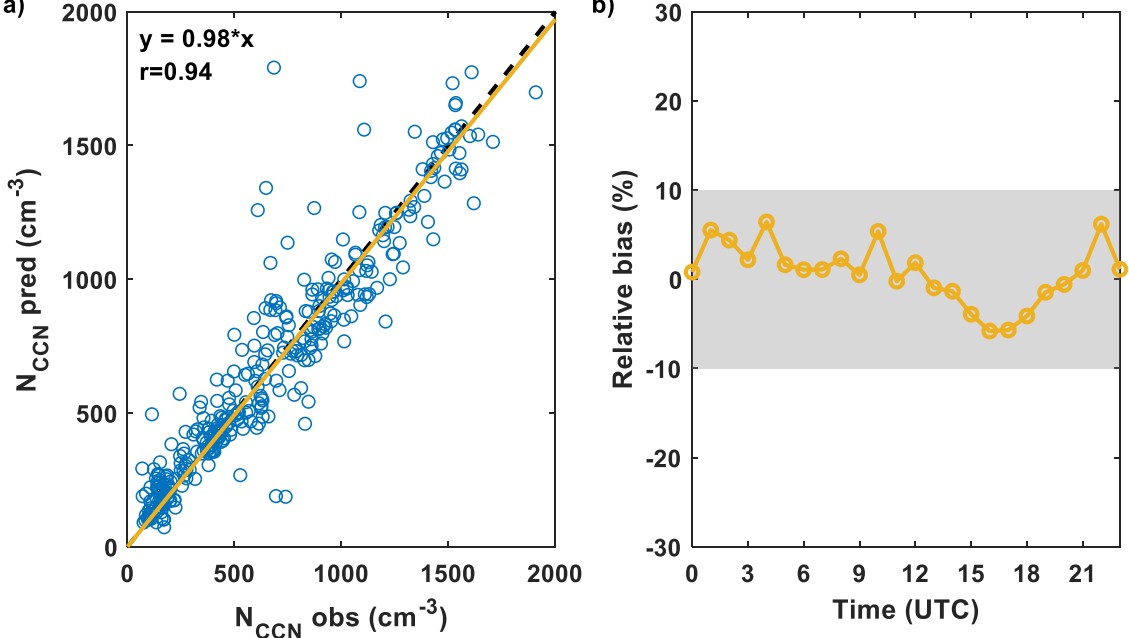


**Figure 9. a) Scatter plot of predicted CCN concentrations (N$_{CCN}$ pred) as a function of observed CCN concentrations (N$_{CCN}$ obs) for the testing sub-dataset (25% of the whole dataset) at SS=0.4% using the neural network approach. The linear regression and Pearson correlation coefficient (r) are also included. b) Median diurnal evolution of the relative bias at SS=0.4% of the neural network model. Also, the grey shaded area represents the ±10% relative bias.**

**5 Summary and conclusions**

We analyzed the influence of $\kappa_{OA}$ on CCN estimations from bulk chemical composition measurements using different OA schemes to describe the overall aerosol hygroscopicity. We investigated the physicochemical properties and CCN activity of the aerosol population at a high-altitude mountain site (SNS station) in the south-eastern Iberian Peninsula, where atmospheric conditions can allow cloud formation.

Our results show the important contribution of OA to the total PM$_1$ mass concentration at SNS where it represents up to the 70% of the PM$_1$. After applying PMF analysis, we have determined that MO-OOA and LO-OOA are the main factors that control both OA and total PM$_1$. During nighttime hours aerosol particles are more aged and hygroscopic with a predominant contribution of MO-OOA and inorganic species. During the morning (6:00-10:00 UTC) the aerosol population starts to be affected by orographic buoyant upward flows of aerosol from the urban area due to mountain-valley breeze regime and PBL

influence. During this time LO-OOA and eBC make a higher relative contribution to the aerosol population, resulting in a decrease in the overall hygroscopicity. The aerosol population properties continue to change during midday hours (11:00-16:00 UTC, highest insolation hours), when the LO-OOA factor and nucleation mode particles exhibit the highest



concentration of the day. This is likely caused by SOA formation through photochemical reactions during NPF events, in combination with other sources such as upslope transport.

The CCN concentration has been estimated by using different OA hygroscopicity schemes based on bulk chemical composition. CCN closure for all OA schemes exhibited slopes and correlation coefficients in the range between 1.08-1.40 and 0.89-0.94, respectively. We find that using a fixed size threshold for CCN activation exhibited a very pronounced diurnal pattern. All OA schemes investigated resulted in similar CCN closure statistics. OA schemes performed better at night (bias between -16% and -6%), when the OA is more oxidized and the aerosol is more aged, than during the day, when the OA is

less oxidized, and the aerosol is more influenced by photochemical and boundary layer processes (bias from 0% to 20%). We also propose a new OA scheme based on the $f_{44}$ parameter, which reflects the oxidation degree of the OA and results in similar overall values as the other OA schemes for the closure slope and correlation coefficient (1.13 and 0.93, respectively). The new OA scheme did improve the closure results for more aged aerosol ($f_{44}>0.25$) which is measured at night, but not during the day (bias values up to 40%) when the aerosol is more complex and $f_{44}$ values are lower. These findings indicate that factors

beyond the bulk $\kappa_{OA}$ characterization must be considered when the aerosol is more complex.

We attribute the observed positive bias of all OA schemes to two main causes. First, the ToF-ACSM provides information of a limited aerosol size range often dominated by accumulation particles which is more affected by inorganic species (Meng et al., 2014; Che et al., 2016, 2017) and, therefore, the real $\kappa_{chem}$ of the whole aerosol population might be overestimated by $\kappa_{chem}$ measured with the ToF-ACSM. In addition, we must consider the effect of the well-known differences in the size ranges

considered between the different instruments in this field campaign (ToF-ACSM, 70-700 nm; SMPS, 12-535 nm and CCNc, none size cutoff). Second, when the aerosol population consists of a complex mixture of particles, which at SNS can be observed during PBL influence conditions, the underlying assumptions for estimating CCN predictions based on internally mixed aerosol particles can introduce an intrinsic bias and $\kappa_{OA}$ assumptions have a secondary role. Moreover, during morning and midday hours related to more complex conditions, the relationship between variables might change over time and can have

a non-linear nature. Therefore, the analytical model approaches presented here cannot explain the CCN changes along the day. The big take-away is that the complexity of the aerosol should be considered when using bulk chemical composition measurements to predict CCN concentrations worldwide .

For that reason, we built a neural network approach which is able to predict CCN concentrations throughout the day. Using four input parameters for the neural network ($N_{80}$, OA/PM$_1$, $f_{44}$ and surface global radiation), we were able to predict accurately

$N_{CCN}$ in all conditions throughout the day (within ±10% relative bias) revealing that this approach was the best for CCN predictions at this complex remote site. It is important to note that the disadvantage of predicting atmospheric parameters using neural networks is that the model is a "black box" which is trained with data of a specific site and can only forecast in that specific site or similar locations. Despite this, it may be possible to use neural networks improve our understanding of global CCN coverage using few aerosol parameters without needing to consider details of aerosol complexity such as mixing state.




**Data availability**

The data used in the manuscript is available from the first author at frejano@ugr.es.

**Author contributions**

FR performed the data harmonization, treatment and formal analysis and wrote the manuscript. AC, MV, JACV and GT carried out the conceptualization and investigation and, together with HL and EA, they carried out a thorough proofreading of the manuscript before obtaining the final version of the manuscript. AA, DP, FJO and LA assisted in the conceptualization. AC, SC, FJG, and GT provided the experimental datasets. All authors contributed to the discussion of the results and provided comments on the paper.

**Competing interests**

The authors declare that they have no conflict of interest.

**Financial support**

This work received support from the European Union's Horizon 2020 research and innovation program through projects ACTRIS.IMP (grant agreement No 871115) and ATMO_ACCESS (grant agreement No 101008004), by the Spanish Ministry of Science and Innovation through projects BioCloud (RTI2018.101154.A.I00) funded by MCIN/AEI/ 10.13039/501100011033 and FEDER "Una manera de hacer Europa", NUCLEUS (PID2021-128757OB-I00) funded by MICIU/AEI/10.13039/501100011033 and FEDER, ELPIS (PID2020-120015RB-100) MCIN/AEI/10.13039/501100011033 and ACTRIS-España RED2022-134824-E. Also, by University of Granada Plan Propio through Visiting Scholars (PPVS2018-04) and Singular Laboratory (AGORA, LS2022-1) programs. Fernando Rejano is funded by Spanish ministry of universities through predoctoral grant FPU19/05340. Andrea Casans is funded by Spanish ministry of research and innovation under the predoctoral program FPI (PRE2019-090827) funded by MCIN/AEI/ 10.13039/501100011033, FSE "El FSE invierte en tu futuro". Elisabeth Andrews acknowledges support from NOAA cooperative agreement NA22OAR4320151.

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
