# Peer review of "CCN estimations at a high-altitude remote site: role of organic aerosol variability and hygroscopicity"

_EGUsphere, 2024_

## Referee Comment (RC2)

The manuscript presents an analysis of a combination of in-situ measurements, including aerosol chemistry and size distribution, to interpret their relationship with the measured cloud condensation nuclei number concentration. All figures are clear and legible, and the paper is very well written. The manuscript presents a detailed methodology section, an overview section including aerosol composition and CCN number concentration, followed by the main part of the discussion that focuses on predicting aerosol hygroscopicity/CCN number concentrations using a number of different approaches, and validating these approaches against the measured CCNC.

These approaches include the conventional $k$-Kohler method, using the particle number and size information and a kappa parameter. The second method calculated aerosol hygroscopicity based on aerosol chemistry measurements; then two subsequent 'schemes' provided detailed organic aerosol hygroscopicity. This later method was determined using results from PMF calculations on the ACSM organic aerosol, which separated the organics into three main factors, More Oxidised Oxidised organic aerosol (MO-OOA), less oxidized OOA (LO-OOA), and hydrocarbon OA (HOA). This was followed by another scheme calculating the organic hygroscopicity using f44 (an indicator of aerosol oxidation state) and then finally a method using neural networks was presented. This method included additional variables; Organic fraction (OA/PM1), particle number concentration (N80), and also solar radiation fluxes.

The initial comparisons showed that the hygroscopicity schemes using more detailed information on organic aerosol oxidation state and its hygroscopicity, resulted in much lower aerosol hygroscopicity (and lower variability) than those calculated using K-Kohler theory. These were illustrated using violin plots and later using scatter plots. The scatter plots allowed the assessment of the range of the goodness of fit. Using a proxy for the oxidation state, f44, alone did not result in any improvement and in all cases there was an overprediction of aerosol hygroscopicity during the night and the opposite during the day. The final comparison using neural networks (including 4 variables) worked very well in comparison to CCN values (@ SS of 4%).

**General comments**

1. It would be very useful to include a figure of the SMPS measurements and eventually to illustrate the contribution of different size modes ($N_{ait}$, $N_{nuc}$, $N_{accum}$) over the period of the study. Using the SMPS we can eventually determine to some extent the mixing state of the aerosol population. One would expect a more externally mixed aerosol population during the time of new particle formation.
2. The work showed that the nighttime aerosol hygroscopicity is underestimated but day time is overestimated. One might assume that nighttime aerosol was more internally mixed and daytime aerosol was more externally mixed based on boundary layer dynamics. Can this be inferred from the SMPS measurements?
3. In Figure 5, the data could also be illustrated by coloring the data points using OA/PM1 or by surface global radiation, or f44. This could provide an introduction into why it would be important to include these variables in the neural network.
4. In the analysis using the neural network, how much weight does each one of the variables carry to assure/maintain the good agreement? Does this provide us with additional information of the chemical/physical processes involved in CCNc prediction?
5. A statement is made that the results between the different hygroscopicity schemes are mostly the same with differences only observed during low hygroscopicity periods (or high HOA

events). It would be helpful to see a time series of the CCN and also of the calculated kappa values.

6. The SMPS and the ACSM are both measuring submicron aerosol. The SMPS is a mobility diameter, the ACSM is an aerodynamic diameter. This should be mentioned or at least convert the SMPS diameter to Aerodynamic when commenting on the comparison of both methods. This is also important for the comment in line 650.

7. *Line 429: The size cut for the CCN measurements can be determined using the aerosol particle loss calculator and information on the aerosol sampling inlets at the site. If the author does not consider that the three instruments (SMPS, ACSM, and CCNC), are measuring the same size range it does not make sense to be compare them, or to use variables from one instrument to predict the measurements from another.*

8. Can these size-cut differences explain the variability of the results, especially during the day when nucleation events occurred?

9. The supersaturations measured in by the CCN range from 0.2 to 0.6, what are the typical supersaturations observed in clouds?

10. Fig. 5 We also see that at low SS (0.2%), NCCN agrees well at low number concentrations and deviates from the line at higher CCN. In previous studies, the impact of Ntotal on NCCN has been shown to have an impact. Do the authors consider that the total NCCN affect the predicted results here?

11. How does the fOA vary over the data points in Fig.5?

**Minor comments:**

Line: 164: NCCN this is the first use of NCCN, please define.

**Aerosol chemistry measurements**

What is the impact of the capture vaporizer on the aerosol mass spectra? The capture vaporizer results in a high degree of fragmentation compared to the standard vaporizer. How does this impact the resolved factors in the PMF results?

Looking at the overall aerosol composition, it would be expected to have an acidic aerosol population? Is this the case? Do you expect some different forms of organics to be present?

It is mentioned that the high concentrations of $SO_4$ measured at the site is from high $SO_2$? Is $SO_2$ measured? Is it expected this $SO_2$ to be transported over long distances or is there local sources. How do these composition compare to other altitude stations.

Being an altitude station, is this station often in cloud? Was the station in cloud during these measurements? Such as around the 6$^{th}$ of June and the 20$^{th}$ of June when the humidity approaches 100%.

Measurements with an AE33 were made. What is the contribution of absorbing aerosols in the other wavelengths compared to those at around 660 nm for BC? Could these increased be a result of other types of aerosols?

In Figure S2: Are error bars available for these plots? Are these differences significant?

Was there a change in wind direction or airmass source during the first and second halves of the field campaign. An analysis of airmass history and back trajectory would be welcome in this work.

Is the site in the free troposphere at night compared to the day time?

In previous work, it was mentioned that NPF is a major source of CCN at this mountain sites. In this work, you also had chemical composition. What were the chemical signatures during these NPF events.

In Figure 3: Having grid lines on the time of day would help to guide the eye of the reader. If this is possible.

Figure 3: Since the magnitude of values between the day and the night are being compared, it would be appreciated to include error bars on these graphs. This would help determine the significance of the differences.

In Figure 2 a time series of aerosol properties is shown which illustrates how the composition changed over time. Were airmass trajectory calculations made for this study?

Line 651: "None size cut" , change to no size cut. Although there is no size cut for this instrument, there is still a size cut implied by the inlets. This can be shown here. In this sentence, a comparison is made with an aerodynamic diameter with a mobility diameter. These are not exactly the same and should be considered.

---

## Author Response (AR1)

**Response to Referee 2- Rejano et al.:**

We thank the reviewer for his/her comments and suggestions that have highly contributed to improve the quality of the manuscript. A point-by-point response to the reviewer's comments is included below. Reviewer's comments are noted in bold. Changes in the manuscript are noted between quotation marks, underlined, and are referred to the corresponding line in the new revised version of the manuscript. In the revised version of the manuscript changes are highlighted in yellow.

**Reviewer #2: The manuscript presents an analysis of a combination of in-situ measurements, including aerosol chemistry and size distribution, to interpret their relationship with the measured cloud condensation nuclei number concentration. All figures are clear and legible, and the paper is very well written. The manuscript presents a detailed methodology section, an overview section including aerosol composition and CCN number concentration, followed by the main part of the discussion that focuses on predicting aerosol hygroscopicity/CCN number concentrations using a number of different approaches, and validating these approaches against the measured CCNC.**

**These approaches include the conventional k-Kohler method, using the particle number and size information and a kappa parameter. The second method calculated aerosol hygroscopicity based on aerosol chemistry measurements; then two subsequent 'schemes' provided detailed organic aerosol hygroscopicity. This later method was determined using results from PMF calculations on the ACSM organic aerosol, which separated the organics into three main factors, More Oxidised Oxygenated organic aerosol (MO-OOA), less oxidized OOA (LO-OOA), and hydrocarbon OA (HOA). This was followed by another scheme calculating the organic hygroscopicity using f44 (an indicator of aerosol oxidation state) and then finally a method using neural networks was presented. This method included additional variables; Organic fraction (OA/PM1), particle number concentration (N80), and also solar radiation fluxes.**

**The initial comparisons showed that the hygroscopicity schemes using more detailed information on organic aerosol oxidation state and its hygroscopicity, resulted in much lower aerosol hygroscopicity (and lower variability) than those calculated using K-Kohler theory. These were illustrated using violin plots and later using scatter plots. The scatter plots allowed the assessment of the range of the goodness of fit. Using a proxy for the oxidation state, f44, alone did not result in any improvement and in all cases there was an overprediction of aerosol hygroscopicity during the night and the opposite during the day. The final comparison using neural networks (including 4 variables) worked very well in comparison to CCN values (@ SS of 4%).**

General comments:

**1. It would be very useful to include a figure of the SMPS measurements and eventually to illustrate the contribution of different size modes (N$_{ait}$, N$_{nuc}$, N$_{accum}$) over the period of the study. Using the SMPS we can eventually determine to some extent the mixing state of the aerosol population. One would expect a more externally mixed aerosol population during the time of new particle formation.**

Concerning the contribution of the different modes to the overall aerosol concentration, Figure R1 shows the contribution of the different size modes during the measurement campaign. The

Aitken mode clearly dominates the total aerosol concentration. There is a strong diel variability in aerosol mode contributions. Although this figure illustrates the strong variability observed during the campaign, we decided not to include it in the main text since the relevant information for this analysis is already included in Figure 3a of the manuscript, which shows the diurnal evolution of each aerosol mode in terms of particle number concentration.

[Figure]

*Figure R1. Contribution to the total number concentration for each aerosol mode (Nucleation, Aitken and Accumulation) over the considered period.*

In terms of inferring the aerosol mixing state from SMPS measurements, we note that the SMPS is a sizing instrument which cannot give conclusive information about the aerosol mixing state. Identifying different aerosol modes in the aerosol size distribution allows us to relate them only to different aerosol types, with different sources or origins. However, it is not possible to determine if the aerosol population is internally or externally mixed using SMPS measurements alone. For example, we cannot distinguish if a unimodal size distribution measured with the SMPS is either an internal mixture of particles or formed by an externally-mixed aerosol population combining different aerosol types with the same size. It has been shown that different combinations of aerosol size distribution and chemical speciation across the aerosol population can coexist with different mixing states (Riemer et al., 2019).

During NPF events at SNS, different aerosol types coexist in the atmosphere (fresh nucleated particles, vertically transported particles, and background particles) and, therefore, we would expect more externally mixed conditions during NPF events. Also, this is consistent with our results showing the highest bias values for the different hygroscopicity schemes during midday hours (when NPF events affect the aerosol population) since for this approach the CCN estimations are based on internally mixed assumptions. Anyway, as we discuss above, from SMPS measurements we cannot determine accurate information about aerosol mixing state.

**2. The work showed that the nighttime aerosol hygroscopicity is underestimated but daytime is overestimated. One might assume that nighttime aerosol was more internally mixed and daytime aerosol was more externally mixed based on boundary layer dynamics. Can this be inferred from the SMPS measurements?**

As mentioned in the question above, it is not possible to determine if the aerosol population is internally or externally mixed with SMPS measurements alone. However, Riemer et al. (2019) state that CCN closure studies provide an indirect way of determining mixing state, since the calculations of CCN concentration are generally sensitive to mixing state assumptions. Since our hygroscopicity schemes are based on internally mixed assumptions, we would expect that

during nighttime the aerosol is more internally mixed since we observed the lowest bias. In contrast, the highest bias is observed during midday hours when we expect more externally mixed particles. Therefore, assessing how good the CCN closure is provides an indirect approach to infer some information about aerosol mixing state which is consistent with the reviewer's hypothesis.

**3. In Figure 5, the data could also be illustrated by coloring the data points using OA/PM1 or by surface global radiation, or f44. This could provide an introduction into why it would be important to include these variables in the neural network.**

We acknowledge the referee's suggestion; however, we think that the discussion presented in the different subsections of the manuscript justify the selection of each variable for the neural network analysis. That said, we have considered this type of color-coding analysis during the preparation of the manuscript and the results were not conclusive enough to include it. For example, Figure R2 shows the results for CCN predictions using OA scheme 3 at SS=0.4% (similar results are observed for the other schemes and SS values) and coloring the data as the reviewer suggested. These figures do not show a clear distribution of data according to the color code, particularly for OA/PM1. For f44 and solar irradiance there some patterns (low f44 and high solar irradiance are both more associated with higher $N_{CCN}$ concentrations). Nonetheless the data points along the 1:1 line indicate a wide range of values of the analyzed variable, limiting the utility of these plots to suggest the influence of these variables (OA/PM1, f44 or solar irradiance) on CCN prediction. Also, we find that the actual Figure 5 in the manuscript is very important to show the amount of data in each pixel of the scatterplot to realize that the majority of datapoints are close to the 1:1 line.

[Figure]

*Figure R2. Scatter plot comparing $N_{CCN}$ predictions for Scheme 3 respect to observation at SS=0.4%. Datapoints of the different panels are color-coded according to different variables: OA fraction (left panel), f44 (central panel) and solar global irradiance (right panel). In each panel the 1:1 line is represented as a black dash line.*

**4. In the analysis using the neural network, how much weight does each one of the variables carry to assure/maintain the good agreement? Does this provide us with additional information of the chemical/physical processes involved in CCNc prediction?**

In a neural network, the weights of the input variables are not predefined but are adjusted during the training process. These weights are determined and adjusted by the backpropagation algorithm with Bayesian regularization. All the variables in the input layer have a priori the same weight, then the backpropagation algorithm adjust the weights according to the training dataset. At the end of the training process, we can analyze the weight of each variable in the activation of all specific hidden neurons and then infer which are the most important variables activating each hidden neuron (which means the highest absolute value for the weights). Figure R3 shows the heatmap of the variable's weights respect to each hidden neuron. We can observe that along the hidden neurons (i.e., across the rows) there is not a clear predominance of one variable. This means that a multivariate relationship involving multiple different physical/chemical processes needs to be used to explain the CCN concentration. However, we

can state that the combination of $N_{80}$ (input 1) and solar global solar irradiance (input 4) show the highest absolute weight values (darkest blue and red colors).

[Figure]

*Figure R3. Heatmap of the weights of the input layer, connecting input variables with hidden neurons*

**5. A statement is made that the results between the different hygroscopicity schemes are mostly the same with differences only observed during low hygroscopicity periods (or high HOA events). It would be helpful to see a time series of the CCN and also of the calculated kappa values.**

In the manuscript, we state that the differences obtained in the kappa parameter between scheme 2 and 3 with respect to scheme 1 are mainly due to the sporadic HOA peaks involving lower hygroscopicity values during the campaign due to the assumption of completely hydrophobic HOA ($k_{HOA}$=0) for scheme 2 (and also scheme 3) (L410-411: *The main difference in the data distribution between both schemes is observed at low hygroscopicity values, which have been identified as periods of higher HOA contribution (i.e., during HOA peak events)*). In general, kappa parameter differences are small because of the low contribution of HOA to OA, while during sporadic high concentration peaks of HOA this difference is enhanced (see Figure R4 top panel).

Figure R4 shows in the top panel the time series of the hygroscopicity parameter difference between schemes 1 and 2 and HOA mass fraction (scheme 3 is omitted for clarity reasons, but similar conclusions than for scheme 2 are drawn). The bottom panel shows the time series of the CCN concentration difference at SS=0.4% between schemes 1 and 2. From Figure R4 we can observe that the hygroscopicity parameter for scheme 2 shows sporadic lower values respect to scheme 1 (higher values of the difference) related to the corresponding HOA mass fraction peaks. The bottom panel of Figure R4 shows the day-to-day variability of the bias in the CCN concentration of each scheme, being the results practically the same

As mentioned before, the day-to-day variability of these variables is not as useful as the diurnal evolution analysis presented in the manuscript in the results discussion. Unfortunately, we cannot say much more than that since these sporadic HOA peaks represent only few points along the campaign and the overall hygroscopicity parameter is mainly dominated by OOA factors.

[Figure]

*Figure R4. Time series of 30-mintes points of hygroscopicity parameter difference for schemes 1 and 2 and HOA mass fraction (top panel) and CCN concentration difference between predictions and observations at SS=0.4% values fpr the same schemes (bottom panel).*

**6. The SMPS and the ACSM are both measuring submicron aerosol. The SMPS is a mobility diameter, the ACSM is an aerodynamic diameter. This should be mentioned or at least convert the SMPS diameter to Aerodynamic when commenting on the comparison of both methods. This is also important for the comment in line 650.**

We agree with the referee comment, and we have clarified in the text this issue since the specific size range for each instrument should be compared using the same definition of diameter.

To clarify this discussion, we have calculated the corresponding mobility diameter of the ToF-ACSM aerodynamic size range ($D_{aero}$ 70-700 nm). Thus, assuming a mean aerosol effective density of 1.47 g/cm$^3$ along the campaign, shape factor of 1 and mean free path of air of 87.8 nm (at 750 hPa corresponding to SNS altitude), we have calculated that the associated mobility diameter range for the ToF-ACSM is around $D_{mob}$ 58-578 nm (SMPS size range is 12-535 nm). This associated mobility diameter range for the ToF-ACSM has been included in the manuscript to address this issue (Lines 657-659).

***7. Line 429: The size cut for the CCN measurements can be determined using the aerosol particle loss calculator and information on the aerosol sampling inlets at the site. If the author does not consider that the three instruments (SMPS, ACSM, and CCNC), are measuring the same size range it does not make sense to be compare them, or to use variables from one instrument to predict the measurements from another.***

We apologize that the statement made in line 429-434 in the originally submitted manuscript, is confusing and we have removed it from the revised manuscript. The remaining sentences now read (lines 439-442 of the revised version):

"It is important to note that the $\kappa_{CCN}$ accounts only for activated particles in the CCNc, whereas $\kappa_{chem}$ accounts only for aerosol particles in the size range allowed by the aerodynamic lens in the ToF-ACSM. Moreover, both methods assume internally mixed particles to estimate the overall $\kappa$, which is an important limitation in the case of externally mixed particles (Wang et al., 2010; Ren et al., 2018; Kulkarni et al., 2023)."

To follow up, we want to note that all instruments are connected to the same main inlet and therefore, the particles losses along the main inlet should be similar for all instruments (diameters of the individual pipes going to each instrument are different depending on the instrument's flow rate). Also, it's important to clarify that the CCNc is always measuring the same size range and it's not SS-dependent as erroneously stated in the previous version of the manuscript. The experimental set-up is in polydisperse mode, which means that the CCNc inlet was connected directly to the inlet without a priori size-selection. However, the kappa parameter derived from the CCNc measurements show a dependence on the SS, and consequently, on the size of the activated particles ($D_{crit}$ for the different SS were, median values, 109, 74 and 60 nm for 0.2, 0.4 and 0.6% SS). Therefore, concerning the statement that the reviewer is referring to, the comparison between the $\kappa_{chem}$ and $\kappa_{CCN}$ depends on the SS. The kappa value derived from the CCNC measurements is not the same as the $\kappa_{chem}$, since the $\kappa_{CCN}$ refers to the effective hygroscopicity that is representative of activated particles. The size of the activated particles depends on the SS, and therefore different $\kappa_{CCN}$ values are obtained for the different SS. According to our estimation of the corresponding mobility diameter range for the ToF-ACSM (58-578 nm), the $D_{crit}$ obtained from CCNc and SMPS always remain within the ToF-ACSM size range. For that reason, all the activated particles (can be assumed as all particles above the $D_{crit}$) should be accounted for by the ToF-ACSM chemical speciation.

**8. Can these size-cut differences explain the variability of the results, especially during the day when nucleation events occurred?**

According to the reviewer´s comment, the size-cut differences could be more important during NPF events since the aerosol concentration in the size range of 12-58 nm, which is not accounted for by the ToF-ACSM, is enhanced. However, in terms of CCN predictions we don't expect a significant influence on our results because particles in the aerosol mobility diameter size range between 12-58 nm barely contribute to CCN concentration at any of the SS values measured by the CCNc.

**9. The supersaturations measured in by the CCN range from 0.2 to 0.6, what are the typical supersaturations observed in clouds?**

Typical supersaturations observed in clouds vary depending on the type of cloud and atmospheric conditions, but they generally fall within the range around 0.2-0.8% (Hudson et al., 2010). Hammer et al. (2014) presented a method to retrieve the so-called effective peak supersaturation $SS_{peak}$, as a measure of the SS at which ambient clouds are formed, at Jungfraujoch station. Their results showed overall median values were 0.22% and 0.41% for southeast and northwest wind conditions, respectively, in agreement with cumulus and shallow clouds. Therefore, the selected SS values are within typical SS observed in clouds.

**10. Fig. 5 We also see that at low SS (0.2%), NCCN agrees well at low number concentrations and deviates from the line at higher CCN. In previous studies, the impact of Ntotal on NCCN**

**has been shown to have an impact. Do the authors consider that the total NCCN affect the predicted results here?**

We agree with the reviewer´s statement that CCN predictions fit better to observation for low CCN concentration values, and it could be due, as we explained in the manuscript, to the fact that low CCN concentration is related to nighttime hours when the aerosol population is more aged (and also possibly more internally mixed) and, therefore, easier to predict. Our CCN predictions tend to overestimate observation during daytime hours when we have a more complex aerosol population. At SNS, high CCN concentration and $N_{tot}$ values coincide with more complex aerosol population and for that reason we observed worse agreement in those conditions. Thus, we cannot say that the over-prediction is only due to a high $N_{tot}$ values. We expect that an internally mixed aerosol population with either high or low $N_{tot}$ values would be well predicted, but we cannot assure that from our experimental data lacking mixing state measurements.

**11. How does the fOA vary over the data points in Fig.5?**

As we explained in question 3, Figure R2 (left panel) shows how $f_{OA}$ (OA/PM$_1$) vary for CCN predictions of OA scheme 3 at SS=0.4%. We observe that the relationship between observed and predicted CCN for both high and low $f_{OA}$ values are close to the 1:1 line. Therefore, changes in the $f_{OA}$ value do not result in a consistent deviation with respect to the 1:1 line. This observation suggests that more complex relation is needed to explain the bias between predicted and measured $N_{CCN}$.

**Minor comments:**

**Line: 164: NCCN this is the first use of NCCN, please define.**

We have modified the manuscript according to reviewer's comment.

**Aerosol chemistry measurements**

**What is the impact of the capture vaporizer on the aerosol mass spectra? The capture vaporizer results in a high degree of fragmentation compared to the standard vaporizer. How does this impact the resolved factors in the PMF results?**
According to an intercomparison of standard vaporizer (SV) - capture vaporizer (CV) shown in Zheng et al. (2020), the main markers for the main sources in SV are still valid for CV, although the relative intensities amongst ions does change. CV source spectra generally present a shift in mass towards lower m/zs especially for the non-oxygenated species, however, in our source apportionment study, the obtained OA sources are well-defined. The HOA spectrum clearly presents the typical signals of the alkyl families ($C_nH^+_{2n-1}$) for a HOA, even if probably more attenuated than they would be for a CV spectrum (see Figure 1, a) b)). For the SOA spectra factors, we got two spectra clearly dominated by the m/z$_{44}$, which allowed a proper identification of them. Their differentiation into two states was based upon the 43-to-44 ratio into a less-oxidized and a more-oxidized oxygenated OA, which were little affected by the different fragmentation pattern.

All in all, we believe that the measurements, analysis, and source apportionment processes were not deeply affected by the use of CV instead of SV and the spectra of the outcoming OA sources are fairly deconvolved and distinct from each other.

**Looking at the overall aerosol composition, it would be expected to have an acidic aerosol population? Is this the case? Do you expect some different forms of organics to be present?**

According to the reviewer´s question, we have calculated the NH4 mass concentration assuming that ammonium neutralizes all sulfate, nitrate, and chloride (NH$_{4\,pred}$) and compared it with NH$_4$ measured by ToF-ACSM (NH$_{4\,measured}$). To estimate NH$_{4\,pred}$, we have used the following equation(Middlebrook et al., 2011):

$$NH_{4\,pred} = M_{NH4} \cdot \left(2 \cdot \frac{SO_4}{M_{SO4}} + \frac{NO_3}{M_{NO3}} + \frac{Cl}{M_{Cl}}\right) \qquad \text{(Equation 1)}$$

These are the molecular weights of each ion:

| Species | $NH_4$ | $SO_4$ | $NO_3$ | $Cl$ |
|---|---|---|---|---|
| $M_X$ (g/mol) | 18 | 96 | 62 | 35.5 |

Figure R6 shows good agreement between the predicted and measured NH4, therefore, the measured aerosol is rather neutralized since most points are close to the 1:1 line. Regarding the presence of organics, the authors do not discount the presence of organo-nitrates or organo-sulphates in the measurements, especially organo-nitrates after SOA night chemistry, but by means of the ToF-ACSM data one cannot detect them.

[Figure]

*Figure R5. Scatter plot of the predicted NH4 mass concentration with respect to the corresponding quantity measured by the ToF-ACSM.*

**It is mentioned that the high concentrations of SO4 measured at the site is from high SO2? Is SO2 measured? Is it expected this SO2 to be transported over long distances or is there local sources. How do these compositions compare to other altitude stations.**

We apologize for this statement since we didn't measure SO2 during this campaign, although the transport of SO2 from Granada urban area is a plausible hypothesis. We have modified the manuscript according to this statement since we cannot determine if SO4 and/or SO2 is transported from long distances or closer areas. The new statement is in lines 274-275 of the revised version:

"*The most abundant inorganic component is SO$_4^{-2}$, which predominates during summertime due to the high temperature and insolation conditions that favor the formation of this compound (Pey et al., 2009; Titos et al., 2014).*"

For comparing the SO4 mass concentration observed at SNS with other mountain sites during the same season, we have to note the altitude of the station and distance to potential sources. At Jungfraujoch, Fröhlich et al. (2013) reported SO4 mass concentration below 1 μg m$^{-3}$ during summer which is similar to our observations (mean value of 0.46 μg m$^{-3}$ at SNS). Higher concentrations of SO4 have been observed at lower altitude stations, for example, Ripoll et al. (2015) reported mean value of 1.9 μg m$^{-3}$ during summer at 1570 m a.s.l and Cai et al. (2021) showed values around 5 μg m$^{-3}$ during summer at 800 m a.s.l. These numbers, suggest there is a clear trend between SO4 mass concentration and the station altitude, although this is likely impacted by proximity to sources as well.

**Being an altitude station, is this station often in cloud? Was the station in cloud during these measurements? Such as around the 6th of June and the 20th of June when the humidity approaches 100%.**

The station is not often in cloud. Since we do not have specific measurements to account for in-cloud periods (i.e., visibility measurements for example), we used a webcam and the measured RH as proxies, and we observed in-cloud conditions in the station during some hours around 5-6 and 20 of June, representing just few datapoints along the campaign. Therefore, the corresponding data were flagged as invalid and not further considered in our analysis. In agreement with the reviewer´s statement, these hours coincide with when the RH reaches values of 100%.

**Measurements with an AE33 were made. What is the contribution of absorbing aerosols in the other wavelengths compared to those at around 660 nm for BC? Could these increased be a result of other types of aerosols?**

First, we want to remark that we have used the 880 nm channel to infer the eBC from AE33 measurements as recommended by Drinovec et al. (2015). In order to figure out how other absorbing aerosols can affect the results; we have calculated the absorption Angstrom exponent (AAE) in two different wavelengths ranges: 370-520 nm and 370-880 nm. The median values along the campaign for AAE in the range of 370-520 nm and 370-880 nm are 1.34 and 1.22, respectively, revealing the predominance of eBC (higher values would have indicated the contribution of other absorbing aerosols). Figure R6 shows the scatter plot of the CCN closure for Scheme 3 at SS=0.4%, with datapoints color-coded by AAE values in the mentioned spectral ranges. The figures show that there is no clear trend on the CCN closure related to AAE values. Therefore, we don't expect a significance influence of other absorbing aerosols in the CCN bias.

[Figure]

*Figure R6. Scatter plot of CCN predictions using Scheme 3 respect observations. Datapoints are color-coded using AAE in spectral range 370-520 nm (left panel) and 370-880 nm (right panel).*

**In Figure S2: Are error bars available for these plots? Are these differences significant?**

According to the reviewer' comment, we have added a shaded area within percentile 25 and 75 of data to Figure S2. Also, we have performed the Wilcoxon rank-sum test (also called Mann-Whitney U) to prove the significance of the difference between both halves of the campaign. The test demonstrated the significance of the differences with a p-value of $10^{-8}$ between both eBC datasets. Although the differences between the two datasets are significant, we cannot be sure that the differences in the OA factors are due to atmospheric conditions.

[Figure]

*Figure S1. Mean diurnal evolution of eBC mass concentration during the first and second half of the campaign. The shaded area is limited by the percentile 25th and 75 of the corresponding data.*

Also, we have modified the manuscript in lines 334-335 of the revised version as follows:

"*During the second half of the campaign, eBC shows a more pronounced diurnal pattern reaching higher concentrations during midday hours compared with the first half of the campaign (Figure S2), however differences between both diurnal patterns are not sufficient to assure that the predominance of each OA factor is related to different atmospheric conditions.*"

**Was there a change in wind direction or airmass source during the first and second halves of the field campaign. An analysis of airmass history and back trajectory would be welcome in this work.**

At the measurement site, the wind direction is predominantly from the west and low wind speeds predominated during the campaign. Figure R7 shows the diurnal pattern of the wind speed (left panel) and direction (right panel) during the two periods. The wind direction was predominantly from the west. For the first half of the campaign, we can observe a clearer diurnal pattern and more influence of other wind directions (lower angles values respect the north direction during nighttime hours). During the second half of the campaign, a more constant wind direction is observed. As we can see in Figure R7 for the wind speed, the diurnal pattern shape is very similar during the whole campaign with higher wind speeds during the evening and lower at the central hours of the day. During the second half of the campaign, we can observe higher values of the wind speed respect to the first half of the campaign. A detailed analysis of the air

masses and wind influence in aerosol composition is shown in Jaén et al. (2023), for the same measurement period.

[Figure]

*Figure R7. Mean diurnal pattern of the wind speed (left panel) and direction (right panel) during the whole campaign, before and after 26th June. The shaded area represents the interquartile range for each variable.*

Despite the significance differences in the wind direction and speed between periods (confirmed by Wilcoxon rank-sum test), CCN predictions are barely affected by the different periods of the campaign. As we can see in Figure R8, we cannot observe a difference in the agreement between $N_{CCN}$ observed and $N_{CCN}$ predicted for scheme 3 (similar discussion for the other OA schemes) when considering the two-time period separately. For the current analysis, we don´t consider that the changes in the wind conditions can explain the bias between measured and predicted CCN concentrations.

[Figure]

*Figure R8. Scatter plot of predicted CCN concentrations ($N_{CCN}$ pred) using OA scheme 3 as a function of observed CCN concentrations ($N_{CCN}$ obs) during the two main periods of the campaign at different SS values.*

**Is the site in the free troposphere at night compared to the daytime?**

The SNS station (2500 m a.s.l) is most of the year in free troposphere, especially during night-time. However, during summer, the mixing layer height can reach the altitude of the station and even higher along the day, mainly during midday and afternoon hours. In Moreira et al. (2020), using remote sensing instrumentation from the valley (Granada, at 680 m asl), the authors showed that the mixing layer height reaches altitudes up to 2000 m a.g.l. (approximately 2680 m a.s.l) during summertime at midday and 500 m a.g.l. (~1180 m a.s.l.) at night.

So, we can affirm that during night the site is most likely in the free troposphere.

**In previous work, it was mentioned that NPF is a major source of CCN at this mountain sites. In this work, you also had chemical composition. What were the chemical signatures during these NPF events.**

We appreciate the reviewer's comment, during this field campaign we also observed NPF events, and a detailed analysis will be presented in a future study (Casans et al., in prep). According to NPF classification done in Casans et al. (in prep), we have obtained the chemical signature for selected NPF and non-NPF events (Figure R9). We can observe that there are differences in the PM1 measurements during events and non-events, with there being a higher contribution of LO-OOA during NPF event compared to non-NPF event days. Also, total PM1 is higher during NPF event than non-events. However, PM1 chemical signature differences could be due to differences in pre-existing particles and in transported particles during the NPF event rather than to the NPF itself since a more efficient vertical transport from lower altitudes has been demonstrated in previous works at SNS during NPF events (Casquero-Vera et al., 2020; Rejano et al., 2021).

[Figure]

*Figure R9. Pie chart of the chemical species for NPF events and non-events during the campaign.*

**In Figure 3: Having grid lines on the time of day would help to guide the eye of the reader. If this is possible.**

We thank the reviewer´s advice and we have updated the Figure 3 in the revised version.

**Figure 3: Since the magnitude of values between the day and the night are being compared, it would be appreciated to include error bars on these graphs. This would help determine the significance of the differences.**

According to the reviewer's comment we have performed a statistical test to determine the significance of the difference between day and night data for all variable shown in Figure 3. To do so, we performed the Wilcoxon rank-sum test (also called Mann-Whitney U) to prove the significance of the difference between two general datasets. The test demonstrated the significance of the difference between day and night data for $N_{CCN}$, $N_{tot}$, $N_{nucl}$, $N_{Aitk}$, $N_{acc}$, $\kappa$ and eBC with a p-value for the test lower than $10^{-10}$, in all cases.

To illustrate this fact, in Figure R10 is shown the diurnal pattern of aerosol particle number concentration and CCN in a linear scale (top panels) and logarithmic scale (bottom panels) with

the interquartile range added. Using the linear scale, we can appreciate the significance of the difference between day and night data.

[Figure]

*Figure R10. Diurnal pattern of particle number concentration (total, nucleation, Aitken and accumulation) and cloud condensation nuclei concentration at different supersaturation values in linear (top panels) and logarithm scales (bottom panels). The shaded area for each variable represents the interquartile range.*

Since the significance of the difference has been proved, we consider that it's not necessary to adapt Figure 3 since including the error bars makes it more difficult to distinguish the diurnal patterns of each variable. According to this issue, we have added some information in lines 357-361 of the revised version of the manuscript:

"*To prove the significance of the difference between night and day hours for all variables presented in Figure 3, we have performed the Wilcoxon rank-sum test (also called Mann-Whitney U). The test demonstrated the significance of the difference between day and night data for all variables (NCCN, Ntot, Nnucl, NAitk, Nacc, κ and chemical species mass concentrations) with a p-value lower than $10^{-10}$, in all cases.*"

**In Figure 2 a time series of aerosol properties is shown which illustrates how the composition changed over time. Were airmass trajectory calculations made for this study?**

As we mentioned before, we performed an airmass trajectory analysis, but it is not used for this study since it does not provide conclusive explanation about CCN prediction. Jaén et al. (2023) presents an overview of airmasses origin and their relationship with $PM_{10}$ chemical composition along this field campaign. Although differences in the chemical composition can be related with different atmospheric scenarios, the connection with the CCN prediction schemes shown in this

study is not conclusive and we prefer to focus on the PM1 chemical composition instead of in the air mass predominance.

**Line 651: "None size cut" , change to no size cut. Although there is no size cut for this instrument, there is still a size cut implied by the inlets. This can be shown here. In this sentence, a comparison is made with an aerodynamic diameter with a mobility diameter. These are not exactly the same and should be considered.**

According to the reviewer comment, we have implemented in the manuscript this suggestion and we have considered the differences in the diameter range specifying when the mobility and aerodynamic diameter is used. Also, we agree that any inlet has an intrinsic cut-off, but a total inlet allows to measure larger particles than $PM_{10}$ and that it´s what we want to remark. Therefore, since we used a total inlet for all the instruments in this field campaign and the CCN counter was measuring directly from the total inlet, we used the term "no size cut" for the CCN counter.

The manuscript has been modified as follows in lines 660-662 of the revised version:

*"…we must consider the effect of the well-known differences in the size ranges considered between the different instruments in this field campaign: CCNc, no size cutoff; SMPS, mobility diameter range of 12-535 nm and ToF-ACSM, aerodynamic diameter range of 70-700 nm (associated with mobility diameter range of 58-578 nm using an effective aerosol density of 1.47 g/cm3 and shape factor of 1)".*

**References:**

Cai, M. F., Liang, B. L., Sun, Q. B., Zhou, S. Z., Yuan, B., Shao, M., Tan, H. B., Xu, Y. S., Ren, L. H., and Zhao, J.: Contribution of New Particle Formation to Cloud Condensation Nuclei Activity and its Controlling Factors in a Mountain Region of Inland China, Journal of Geophysical Research: Atmospheres, 126, e2020JD034302, https://doi.org/10.1029/2020JD034302, 2021.

Casquero-Vera, J. A., Lyamani, H., Dada, L., Hakala, S., Paasonen, P., Román, R., Fraile, R., Petäjä, T., Olmo-Reyes, F. J., and Alados-Arboledas, L.: New particle formation at urban and high-altitude remote sites in the south-eastern Iberian Peninsula, Atmos Chem Phys, 20, 14253–14271, https://doi.org/10.5194/acp-20-14253-2020, 2020.

Drinovec, L., Močnik, G., Zotter, P., Prévôt, A. S. H., Ruckstuhl, C., Coz, E., Rupakheti, M., Sciare, J., Müller, T., Wiedensohler, A., and Hansen, A. D. A.: The "dual-spot" Aethalometer: An improved measurement of aerosol black carbon with real-time loading compensation, Atmos Meas Tech, 8, 1965–1979, https://doi.org/10.5194/amt-8-1965-2015, 2015.

Fröhlich, R., Cubison, M. J., Slowik, J. G., Bukowiecki, N., Prévôt, A. S. H., Baltensperger, U., Schneider, J., Kimmel, J. R., Gonin, M., Rohner, U., Worsnop, D. R., and Jayne, J. T.: The ToF-ACSM: A portable aerosol chemical speciation monitor with TOFMS detection, Atmos Meas Tech, 6, 3225–3241, https://doi.org/10.5194/amt-6-3225-2013, 2013.

Hammer, E., Gysel, M., Roberts, G. C., Elias, T., Hofer, J., Hoyle, C. R., Bukowiecki, N., Dupont, J. C., Burnet, F., Baltensperger, U., and Weingartner, E.: Size-dependent particle activation properties in fog during the ParisFog 2012/13 field campaign, Atmos Chem Phys, 14, 10517–10533, https://doi.org/10.5194/acp-14-10517-2014, 2014.

Hudson, J. G., S. Noble, and V. Jha (2010), Stratus cloud supersaturations, Geophys. Res. Lett., 37, L21813, doi:10.1029/2010GL045197.

Jaén, C., Titos, G., Castillo, S., Casans, A., Rejano, F., Cazorla, A., Herrero, J., Alados-Arboledas, L., Grimalt, J. O., and van Drooge, B. L.: Diurnal source apportionment of organic and inorganic atmospheric particulate matter at a high-altitude mountain site under summer conditions (Sierra Nevada; Spain), Science of the Total Environment, 905, https://doi.org/10.1016/j.scitotenv.2023.167178, 2023.

Middlebrook, A. M., Bahreini, R., Jimenez, J. L., & Canagaratna, M. R: Evaluation of Composition-Dependent Collection Efficiencies for the Aerodyne Aerosol Mass Spectrometer using Field Data. Aerosol Science and Technology, 46(3), 258–271. https://doi.org/10.1080/02786826.2011.620041, 2011.

Moreira, G. de A., Guerrero-Rascado, J. L., Bravo-Aranda, J. A., Foyo-Moreno, I., Cazorla, A., Alados, I., Lyamani, H., Landulfo, E., and Alados-Arboledas, L.: Study of the planetary boundary layer height in an urban environment using a combination of microwave radiometer and ceilometer, Atmos Res, 240, 104932, https://doi.org/10.1016/j.atmosres.2020.104932, 2020.

Rejano, F., Titos, G., Casquero-Vera, J. A., Lyamani, H., Andrews, E., Sheridan, P., Cazorla, A., Castillo, S., Alados-Arboledas, L., and Olmo, F. J.: Activation properties of aerosol particles as cloud condensation nuclei at urban and high-altitude remote sites in southern Europe, Science of the Total Environment, 762, https://doi.org/10.1016/j.scitotenv.2020.143100, 2021.

Riemer, N., Ault, A. P., West, M., Craig, R. L., and Curtis, J. H.: Aerosol Mixing State: Measurements, Modeling, and Impacts, Reviews of Geophysics, 57, 187–249, https://doi.org/10.1029/2018RG000615, 2019.

Ripoll, A., Minguillón, M. C., Pey, J., Jimenez, J. L., Day, D. A., Sosedova, Y., Canonaco, F., Prévôt, A. S. H., Querol, X., and Alastuey, A.: Long-term real-time chemical characterization of submicron aerosols at Montsec (southern Pyrenees, 1570 m a.s.l.), Atmos Chem Phys, 15, 2935–2951, https://doi.org/10.5194/acp-15-2935-2015, 2015.

Zheng, Y., Cheng, X., Liao, K., Li, Y., Li, Y. J., Huang, R.-J., Hu, W., Liu, Y., Zhu, T., Chen, S., Zeng, L., Worsnop, D. R., and Chen, Q.: Characterization of anthropogenic organic aerosols by TOF-ACSM with the new capture vaporizer, Atmos. Meas. Tech., 13, 2457–2472, https://doi.org/10.5194/amt-13-2457-2020, 2020.

**Response to Referee 3- Rejano et al.:**

We thank the reviewer for his/her comments and suggestions that have highly contributed to improve the quality of the manuscript. A point-by-point response to the reviewer's comments is included below. Reviewer's comments are noted in bold. Changes in the manuscript are noted between quotation marks, underlined and are referred to the corresponding line in the new revised version of the manuscript.

**Reviewer #3: This study deals with the complex dependence of CCN activity on the hygroscopicity of organic aerosols. Since CCN activity determines the indirect effect of aerosols on radiative forcing, the subject is of great interest. Using chemical and size distribution measurements, the authors study the role of OA hygroscopicity on CCN activity, for different atmospheric conditions. The positive matrix factorisation method was used to recover the relative contribution of OA with different oxidation levels, showing that medium and low oxidised OA are predominant, and that their contribution varies as a function of the vertical transport of PBL to the site. The originality of this study lies in the use of a neural network model to predict the amount of CCN using aerosol size distribution data, the fraction of OA and a factor of PMF, and radiation. This innovative tool gave the best results compared with assumptions about global chemical composition. The authors conclude by stressing the importance of taking into account the complexity of the aerosol and in particular its internal/external mixing. The manuscript is well structured and well written. The conclusion is clear and the message to be retained is correctly underlined. I consider that the manuscript can be published, after minor revision and responses to the following points:**

**Specific comments**

**1. Please add the equation which links κ, Dcrit and SS at the CCN activation in section 3.2 (κ -Kohler theory). You could then cite it after (eg. L 435).**

We included the aforementioned equation in the manuscript and cited it accordingly.

**2. L 281. Could you explain why the eBC increase starts earlier than the inorganics and OA increase during the day?**

It's something that we are not able to give a conclusive explanation of this fact based on our measurements. However, considering all the possibilities to explain this fact we point out different issues:

The eBC particles are considered primary aerosol particles, while IA and OA are mainly dominated by secondary particles. For that reason, we can expect some delay between the increase of these species. In Figure 3 we can observe that eBC starts to increase at the same time as nucleation mode concentration. Since nucleation mode particles are difficult to form during NPF events at night and the eBC increase coincides with the nucleation mode increase, we can assume that there is some influence of local primary emission. On the other hand, the IA and OA increase during morning hours is caused by vertical transport. Therefore, IA and OA increase may take longer time to be observed.

We have modified the manuscript to clarify this discussion in lines 284-292:

*"Based on these diurnal patterns, inorganic species are most likely transported from the Granada urban area due to upslope mountain breezes and the increase of the PBL height during daytime. OA also increased at midday, but the increase is sharper, reaching a maximum between 12:00-*

*16:00 UTC. OA exhibits a larger increase in concentration at midday hours compared to the other species (Figure 1b), which might suggest the influence of upslope transport but also additional sources of OA in the vicinity of the measurement site (such as local emissions or secondary processes as nucleation). Finally, eBC mass concentration increased more gradually, starting at 3:00 UTC and reaching a maximum at 11:00 UTC. The earlier increase of eBC with respect to IA and OA species might be related to some local primary emissions during the early morning, although most of the eBC observed at SNS is due to upslope transport at midday (Rejano et al., 2021)."*

**3. L 324. What are the wind direction and speed like during the two identified periods ?**

At the measurement site, the wind direction is predominantly from the west and low wind speeds predominated during the campaign. Figure R1 shows the diurnal pattern of the wind speed (left panel) and direction (right panel) during the two periods. The wind direction was predominantly from the west. For the first half of the campaign, we can observe a clearer diurnal pattern and more influence of other wind directions (lower angles values respect the north direction during nighttime hours). During the second half of the campaign, a more constant wind direction is observed. As we can see in Figure R1 for the wind speed, the diurnal pattern shape is very similar during the whole campaign with higher wind speeds during the evening and lower at the central hours of the day. During the second half of the campaign, we can observe higher values of the wind speed respect to the first half of the campaign. A detailed analysis of the air masses and wind influence in aerosol composition is shown in Jaén et al. (2023), for the same measurement period.

[Figure]

*Figure R1. Mean diurnal pattern of the wind speed (left panel) and direction (right panel) during the whole campaign, before and after 26th June. The shaded area represents the interquartile range for each variable.*

Despite the significance differences in the wind direction and speed between periods (confirmed by Wilcoxon rank-sum test), CCN predictions are barely affected by the different periods of the campaign. As we can see in Figure R2, we cannot observe a difference in the agreement between $N_{CCN}$ observed and $N_{CCN}$ predicted when considering the two-time period separately For the current analysis, we don´t consider that the changes in the wind conditions can explain the bias between measured and predicted CCN concentrations.

[Figure]

*Figure R2. Scatter plot of predicted CCN concentrations (N$_{CCN}$ pred) using OA scheme 3 as a function of observed CCN concentrations (N$_{CCN}$ obs) during the two main periods of the campaign at different SS values.*

**L 365. What about the solar radiation diurnal profile role in the photochemical oxidation ?**

We only mentioned additional factors that could contribute to photochemical oxidation, but we didn't mention solar radiation, that is the main factor that control these reactions. Therefore, we have added the solar irradiance diurnal pattern to Figure S3b (see below) and have updated the manuscript as follows:

Line 374-375: "*by SOA formation linked to photochemical oxidation induced by high solar irradiance values and high concentration of O₃ and NOx (Figure S3a) together with high temperatures (Figure S3b)*".

[Figure]

*Figure S2. Mean diurnal evolution of a) NOx (left Y axis) and O3 concentration (right Y axis) and b) temperature (left Y axis) and solar global irradiance (right Y axis) along the campaign.*

Also, we have corrected the manuscript according to the actual physical variable measured by the pyranometer during the campaign which is the global solar irradiance instead of total radiation.

**L 413. What do you mean by "Time-dependent" ? If I've well understood, you described previously that in Scheme 3 specific κ values for LO-OOA and MO-OOA have been used to**

take into account their relative contribution at SNS, but these values are fixed, and not varying as a function of the time. Could you please clarify this ?

According to reviewer's comment, we want to clarify that for Scheme 3 the calculation of $\kappa_{OA}$ is obtained from the relative contribution of each OA factor. It means that $\kappa_{OA}$ value changes over time since it is calculated including the OA temporal variability (each OA factor has a different hygroscopicity and each OA factor has a different contribution to the overall kappa during the day). In contrast, the overall $\kappa$ computation of scheme 1 does not include the OA variability since that scheme assumes a constant $\kappa_{OA}$ of 0.1. Also, for Scheme 2 we can state almost the same since the only difference is including the time variability of HOA in the overall $\kappa$ calculation, which represents the smallest fraction of OA. We acknowledge that the term "time-dependent $\kappa_{OA}$" can be confusing and for that reason we have modified the manuscript in lines 421-423 to clarify this statement:

"*Scheme 3 exhibits a clearly different data distribution compared to schemes 1 and 2 due to the assumption of different $\kappa$ values for the LO-OOA and MO-OOA factors.*"

**L 443. Could you add the Dcrit values for SS= 0.2 and 0.6% to compare it with Dcrit=72 nm?**

We agree with the useful reviewer comment, and we have added the mean Dcrit value at 0.2% and 0.6%. The text is modified in lines 451-452 as follows:

"*In this case, the predicted $N_{CCN}$ values overestimate the measurements at low SS (mean $D_{crit}$ is 111±21 nm) and underestimate the measurements at high SS level (mean $D_{crit}$ is 58±16 nm). At SS=0.4% the mean $D_{crit}$ is 72±18 nm and…*"

**L 451 to 461. Did you observe a difference in the agreement between NCCN observed and NCCN calculated when considering the two-time period separately (before and after June 26th )?**

We have observed some difference in some aerosol properties and atmospheric conditions in the two periods, but we have not observed a clear impact in the CCN closure agreement. To support this statement, we show the scatter plot of predicted $N_{CCN}$ using OA scheme 3 respect to observed $N_{CCN}$ for each time period for all three SS considered (Figure R2, discussed above). There are hardly any differences between the two measurement periods of the campaign in terms of CCN closure agreement. For that reason, we decided during the manuscript preparation not to focus on the differences between the two periods and focus on the importance of chemical composition and other variables (N80, radiation) which are measured at high-time resolution and provide better insight into the CCN predictions.

**Technical corrections:**

**Please check the units notation in the text and the figures (eg. μg m-3 instead of μg/m3)**

Thanks for the recommendation, we have corrected this issue along the manuscript.

**Figure 4 : please add a note on the shaded areas representing the PDF of each variable**

We thank the reviewer´s comment and we have updated the Figure 4 caption as follows:

"*Figure 4. Violin plot of $\kappa$ distribution data for the chemical schemes ($\kappa_{chem}$) and the CCN calculation at different SS values ($\kappa_{CCN}$). The boxes represent the interquartile distance, and the*

*asterisk is the mean value. The shaded area for each variable represents the probability density function (PDF)".*

**L 489. "is associated with"**

Done

**Figure 6: Please add a) and b) on the different panels.**

We have modified Figure according to the reviewer's comment.

**L 513.: please give the values used for κIA and κBC**

According to that comment, we don´t use a specific value for κ$_{IA}$, it is derived from ToF-ACSM measurements and the hygroscopicity of each inorganic compound provided in the Table S1. The κ$_{BC}$ is always assumed as 0. We have modified the manuscript to avoid misunderstanding. See the new text in lines 523-524 in the revised version:

*"In this sub-section we calculate $\kappa_{OA}$ from Equation 5 using the overall aerosol hygroscopicity as $\kappa_{CCN}$, $\kappa_{IA}$ obtained from ToF-ACSM measurements assuming specific hygroscopicity values for each inorganic compound shown in Table S1 and $\kappa_{BC}$ assumed as 0 (Cerully et al., 2015; Kuang et al., 2020; Thalman et al., 2017):"*

**Figure 8a. : Could you sort data by f44 values such that the points with the lowest f44 are plotted in first and the points with the highest f44 and plotted over (ie. yellow points overlapping the blue points) ?**

According to the reviewer's comment, Figure R3 shows the implementation of his/her suggestion. Data are sorted according to the f44 values if they are below/above f44=0.25. Then, we plot small markers over datapoints with f44 value below 0.25 which are used in the filtered fit. Anyway, we prefer not to include this update in the revised version since **we prefer to include** the color-coded scale in this figure.

[Figure]

*Figure R3. a) Scatter plot of predicted CCN concentrations (NCCN pred) as a function of observed CCN concentrations (NCCN obs) using the OA scheme 4. Datapoints with f44>0.25 are represented as grey triangles and f44<0.25 as small blue dots. The black dash line represents the 1:1 line. The linear equation and Pearson correlation coefficient (r) are also included for all data and for filtered data in parenthesis. The yellow solid and dash lines represent the linear regression of all and filtered data, respectively. b) Median diurnal evolution of the relative bias at SS=0.4% of the OA scheme 4 (left Y axis) and f44 (right Y axis). The grey shaded area represents the ±10% relative bias. The red shaded area represents the relative bias range for the other OA schemes shown in Figure 6b.*

**L 588. Slopes and correlation coefficients don't correspond to the ones in Fig. 9a.**

Thanks to the reviewer´s comment we have corrected this mistake in the manuscript and updated it with the correct values presented in Figure 9a.

**L 603. R2 = 0.88 or 0.94 (cf. Fig. 9a) ?**

We again thank the reviewer; the R value is 0.94 and we have corrected the manuscript according.

References

Jaén, C., Titos, G., Castillo, S., Casans, A., Rejano, F., Cazorla, A., Herrero, J., Alados-Arboledas, L., Grimalt, J. O., and van Drooge, B. L.: Diurnal source apportionment of organic and inorganic atmospheric particulate matter at a high-altitude mountain site under summer conditions (Sierra Nevada; Spain), Science of the Total Environment, 905, https://doi.org/10.1016/j.scitotenv.2023.167178, 2023.

---

## Author Response (AR2)

**Editor decision: Publish subject to minor revisions (review by editor)**:

We thank the Editor for reviewing the manuscript and for her comments to improve the quality of the manuscript. A point-by-point response to each comment is included below. Editor's comments are noted in bold. Changes in the manuscript are noted between quotation marks, underlined, and are referred to the corresponding line in the new revised version of the manuscript.

**Dear Authors,**

**Thank you for the revised manuscript which has clearly improved thanks to reviewer's excellent suggestions.**

**As a final note, I would encourage the authors to reconsider incorporating some of the reviewer's suggestions better into the manuscript, particularly:**

**1. Reply on the comment concerning Figure 8a. I understand this comment differently, as a request to plot the points in the figure in the order from the smallest f44 towards the highest f44 (instead of ordering them with time), such that the highest values would be clearly visible and not covered by blue points. So please check this once more.**

Sorry for misunderstanding the reviewer's comment, we have now considered the suggestion in the correct way, and we have updated Figure 8a accordingly. Also, the marker of the filtered data (small black dots in the previous version of the manuscript) has been removed for better visualization of the datapoints' colors.

[Figure]

**Figure 8. a)** Scatter plot of predicted CCN concentrations ($N_{CCN}$ pred) as a function of observed CCN concentrations ($N_{CCN}$ obs) using the OA scheme 4. Datapoints are colored-coded by the corresponding $f_{44}$ value. The black dash line represents the 1:1 line. The linear equation and Pearson correlation coefficient (r) are also included for all data and for filtered data in parenthesis (datapoints with $f_{44}>0.25$). The yellow solid and dash lines represent the linear regression of all and filtered data, respectively. **b)** Median diurnal evolution of the relative bias at SS=0.4% of the OA scheme 4 (left Y axis) and $f_{44}$ (right Y axis). The grey shaded area represents the ±10% relative bias. The red shaded area represents the relative bias range for the other OA schemes shown in Figure 6b.

**2. Adding possibly discussion of the reviewer's concerns (particularly the points raised by reviewer #2), such as the impact of external/internal aerosol mixtures on the comparison results. If this is not possible to evaluate quantitatively, could it still be addressed as an uncertainty/limitation of the study?**

Following editor's suggestion, we have revised the whole manuscript in order to remark (in the results and conclusion sections) the limitation of the study to evaluate the impact of the aerosol mixing state on the prediction models' performance. However, we want to remark that the discussion of the results in the manuscript does not quantitatively address this issue since it is not possible with the measurements available in this study.

**In lines 450-454:** "Moreover, both methods assume internally mixed particles to estimate the overall $\kappa$, which is an important limitation in the case of externally mixed particles (Wang et al., 2010; Ren et al., 2018; Kulkarni et al., 2023). In this study, we would expect that during midday hours at SNS the aerosol population would be more externally mixed due to the influence of NPF events and vertical transport of particles; however, with the instrumentation and methods used here we cannot give conclusive information about aerosol mixing state and its impact on the different scheme's performance.".

**In lines 604-605:** "However, the analytical methods used in this study have limitations and do not consider aerosol mixing state information. For that reason, the next section will explore a non-analytical approach with the aim of improving the CCN prediction throughout the day."

**In lines 675-680:** "Second, when the aerosol population consists of a complex mixture of particles, which at SNS can be observed during PBL influence conditions, the underlying assumptions for estimating CCN concentrations based on internally mixed aerosol particles can introduce an intrinsic bias and $\kappa_{OA}$ assumptions have a secondary role. Since the methodology used here is limited and cannot directly infer the mixing state of the aerosol population, we are not able to quantify this effect on the CCN predictions. Moreover, during morning and midday hours related to more complex conditions, the relationship between variables might change over time and can have a non-linear nature."

**3. Adding possibly supporting figure (e.g. in supplement) to strengthen the statement in lines L410-411: "The main difference in the data distribution between both schemes is observed at low hygroscopicity values, which have been identified as periods of higher HOA contribution (i.e., during HOA peak events). ", as was suggested.**

We have added the following figure to the supplement of the manuscript to support the statement in lines L410-411 as suggested by the reviewer and the Editor. Lines 430-431 of the revised version have been updated as follows:

"The main difference in the data distribution between both schemes is observed at low hygroscopicity values, which have been identified as periods of higher HOA contribution (i.e., during HOA peak events) as can be observed in Figure S5."

[Figure]

**Figure S5. Time series along the field campaign of hygroscopicity parameter difference between both schemes (left Y-axis) and HOA mass fraction (right Y-axis).**

**4. clarifying the wind conditions, since this question was raised by both reviewers.**

To clarify the discussion about the wind conditions during the campaign, we have added the following sentences in lines 330-340 of the revised version of the manuscript and the figure attached below in the supplement (Figure S2):

"Figure S1 shows the timeseries of meteorological variables (temperature, pressure, and relative humidity) during the campaign. The second half of the campaign is characterized by higher temperatures, higher pressure and lower relative humidity compared to the first half. In addition, we observed that the diurnal pattern of wind speed is very similar during the whole campaign (with higher wind speeds during the evening and lower at the central hours of the day), but the second half of the campaign shows higher values of the wind speed respect to the first half of the campaign (Figure S2). The wind direction was also predominantly from the west for the whole campaign, but we observed that the first half of the campaign shows a more pronounced diurnal pattern and more influence of other wind directions (Figure S2). During the second half of the campaign, a more constant wind direction is observed suggesting a continuous transport of aerosol from the valley to the mountain. The significant difference in the meteorological conditions between both halves of the campaign can be associated to a more efficient transport of aerosol from the valley to the mountain during the second half of the campaign, which would involve differences in the aerosol physico-chemical properties, and in particular in the predominance of the OA factors (50% contribution of LO-OOA and 48% of MO-OOA during the second half compared to 26% and 72% during the first half). A detailed analysis of the air masses and wind influence in aerosol composition during BioCloud campaign is shown in Jaén et al. (2023)."

[Figure]

**Figure S2. Mean diurnal pattern of the wind speed (left panel) and direction (right panel) during the whole campaign, before and after 26th June. The shaded area represents the interquartile range for each variable.**